# OF-CEAS laser spectroscopy to measure water isotopes in dry environments: example of application in Antarctica

Thomas Lauwers[1], Elise Fourré[1], Olivier Jossoud[1], Daniele Romanini[3], Frédéric Prié[1], Giordano Nitti[1], Mathieu Casado[1], Kévin Jaulin[2], Markus Miltner [2,3], Morgane Farradèche[1], Valérie Masson – Delmotte[1],

Amaëlle Landais[1]

[1]Laboratoire des Sciences du Climat et de l'Environnement – IPSL, UMR 8212, CEA-CNRS-UVSQ, Paris-Saclay University, Gif-sur-Yvette, France
[2]AP2E, 110 av. Galilée, 13290 Aix en Provence, France
[3]Université Grenoble Alpes, CNRS, LIPhy UMR 5588, 38041 Grenoble, France

*Correspondence to*: Thomas Lauwers (thomas.lauwers@lsce.ipsl.fr)

**Abstract.** Water vapour isotopes are important tools to better understand processes governing the atmospheric hydrological cycle. Their measurement in polar regions is crucial to improve the interpretation of water isotopic records in ice cores. *In situ* water vapour isotopic monitoring remains challenging, especially in dry places of the East Antarctic plateau where water mixing ratio can be as low as 10 ppm. We present in this article new commercial laser spectrometers based on the optical

feedback – cavity enhanced absorption spectroscopy (OF-CEAS) technique, adapted for water vapour isotopic measurement in dry regions. We characterize a first instrument adapted for Antarctic coastal monitoring with an optical cavity finesse of 64 000 (ringdown time of 54 µs), installed at Dumont d'Urville station during the summer campaign 2022-2023, and a second instrument with a high finesse of 116 000 (98 µs ringdown), to be deployed inland East Antarctica. With a drift calibration every 48 hours, the stability demonstrated by the high finesse instrument allows to study isotopic diurnal cycles down to 10

ppm humidity for $\delta D$ and 100 ppm for $\delta^{18}O$.

## Introduction

Water vapour stable isotope monitoring (mainly $H_2^{16}O$, $H_2^{18}O$ and $HD^{16}O$) in the atmosphere helps to understand a number of processes governing the atmospheric water cycle (Galewsky et al., 2016), such as phase change (Merlivat and Nief, 1967;

Benetti et al., 2018; Hughes et al., 2021), transport (Bonne et al., 2020), or mixing of air masses. Until the 1990s, the first techniques for water vapour isotopic composition monitoring relied on sampling with cryogenic traps and subsequent mass spectrometry measurements (Angert et al., 2008), but it was time consuming and not easy to implement in a broad variety of environments.

Today, laser spectrometers are a solution for *in situ* continuous measurements (Gupta et al., 2009; Landais et al., 2024). Isotope

analysers use near infrared laser diodes and most of them are based either on the cavity ring-down spectroscopy technique (CRDS) or the cavity enhanced absorption spectroscopy technique (CEAS). The CRDS method, which is commonly implemented by the Picarro company, achieves a high stability through the measurement of the photon lifetime inside the optical cavity instead of the direct absorbed light. Those instruments are robust and adapted for field measurement. A broad number of studies used water vapour stable isotopes to document the evolution of the atmospheric water cycle over synoptic

events (e.g., cold fronts, cyclones) (Aemisegger et al., 2015; Bhattacharya et al., 2022; Tremoy et al., 2014) or to understand processes within the water cycle (e.g., evaporation over the ocean) (Benetti et al., 2015). Instruments are no longer only installed in observatory stations but can be found on board boats (Thurnherr et al., 2020) or aircrafts (Henze et al., 2022). An increasing number of studies are also now devoted to the study of the atmospheric water cycle in the polar regions with the objective to document either the atmospheric dynamics (e.g., atmospheric rivers, synoptic events, influence of katabatic winds)

(Bonne et al., 2014; Bréant et al., 2019; Kopec et al., 2014; Leroy-Dos Santos et al., 2021, 2023) or the exchange between snow and water vapour at the surface of the ice sheets (Casado et al., 2016; Ritter et al., 2016; Wahl et al., 2021). Those last studies are essential to interpret the water isotopic records in ice cores, which are not only driven by temperature and condensation along the transportation of water vapour from the evaporative to the polar regions but also influenced by

equilibrium / diffusive processes in the upper snow (Dietrich et al., 2023). However, CRDS struggles to properly measure the isotopic composition in very dry conditions (water mixing ratio below 500 ppm) (Leroy-Dos Santos et al., 2021), that can be encountered in polar regions or at altitude, so that isotopic processes in key regions like inland Antarctica can only be documented during summer (Casado et al., 2016; Ritter et al., 2016).

To overcome this limitation, we present in this article instruments based on an alternative technique called OF-CEAS, which combines the CEAS method and an optical feedback (OF) from a V-shaped cavity. This allows us to stabilise the laser emission frequency by locking it successively to the multiple cavity resonances (Morville et al., 2014; Romanini et al., 2014). This provides efficient cavity injection and low noise cavity output from all resonances across the laser scan. The maxima of these resonances provide directly the cavity enhanced spectrum, converted to an absolute absorption scale using a ring-down produced by shutting off the laser at the last resonance in the laser scan (Romanini et al., 2014). This technique was first implemented for water vapour isotope analysis with a laboratory prototype under stable working conditions (Landsberg et al., 2014), but never successfully deployed in the field for extended periods. In this paper, we present the performance obtained with new commercial OF-CEAS analysers, developed in collaboration with the AP2E company (ProCeas®) and specifically designed to measure water vapour isotopes in a very dry environment. After a brief description of the laser spectrometer and the auxiliary calibration instrument, we present the analyser stability, its water mixing ratio response, and its accuracy and precision in dry conditions. We finally propose a calibration procedure adapted for continuous water vapour isotope monitoring using OF-CEAS instruments, and discuss the instrumental performance compared to already available commercial instruments manufactured by Picarro.

## 1 Instrumental development

### 1.1 OF – CEAS spectrometer

The AP2E ProCeas® analysers presented in this study are based on the OF-CEAS technique originally implemented in laboratory prototypes (Landsberg et al., 2014; Lechevallier et al., 2019). To adapt the analyser for field measurement, AP2E made a number of improvements in terms of robustness and instrumental stability, mainly by designing new custom mirrors and laser mounts and by implementing a high precision temperature and pressure regulation. For a complete description of the ProCeas® system, the reader may refer to the recent article of Piel et al. (2024) describing the OF-CEAS spectrometer used for atmospheric $O_2$ isotopic measurement.

The OF-CEAS spectrometers for the measurement of water isotopologues use a distributed feedback laser source centred around 1 389 nm to target the three water absorption lines of HDO (7 200.3023 $cm^{-1}$), $H_2^{16}O$ (7 200.1335 $cm^{-1}$) and $H_2^{18}O$ (7 199.9614 $cm^{-1}$). As shown in the spectrum in Figure 1, the absorption lines of interest (blue line) can be affected by the presence of methane (grey line) and strong absorption lines of water located outside the spectral window.

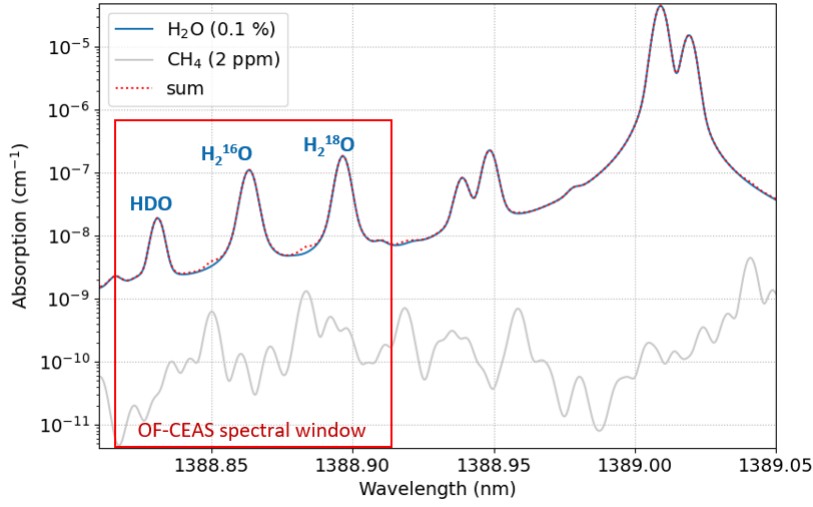


**Figure 1 : Absorption spectrum of the three target isotopologues HDO, H$_2^{16}$O and H$_2^{18}$O calculated from the 2020 HITRAN database. The total absorption spectrum is plotted with the red dotted line, considering 0.1% of water vapour (blue line) and 2 ppm of methane (grey line). The red rectangle indicates the OF-CEAS spectral window by current tuning of a 1 389 nm distributed feedback laser diode.**

The centring of the spectral window is achieved by tuning the temperature of the laser source whereas the fast wavelength scan is performed by tuning the laser current.

With a cavity length of about 40 cm resulting in a free spectral range (FSR) of 188 MHz, the wavelength range of interest contains 80 resonance modes, as shown in Figure 2 (blue dots). The spectrum was obtained after a long injection of dry nitrogen, resulting in a minimal absolute humidity of 3 ppm. The residuals (difference between the fitted and acquired

spectrum) are shown by the yellow line. The spectral fitting is performed using Voigt profiles for the water and methane absorption lines and an additional quadratic baseline to account for background absorption losses. To adjust the fitting, the physical spectroscopic values of water and methane are first retrieved from the Hitran database (mainly the relative position of the peaks, intensities, Gaussian and Lorentzian width), and used as initial parameters. Then, the parameters are empirically tuned to obtain the smallest and flattest residuals for a wide range of different gas matrices (pure nitrogen, atmospheric dry air

and finally synthetic air with a low water content). For example, a symmetric shape of the residuals around the peaks such as a M-shape or a W-shape would indicate an incorrect width, while an asymmetric shape would indicate a non-optimized peak position. The resulting residuals after optimization show a uniform repartition, with a peak-to-peak value of $1.2 \times 10^{-10}$ cm$^{-1}$, as shown in Figure 2.

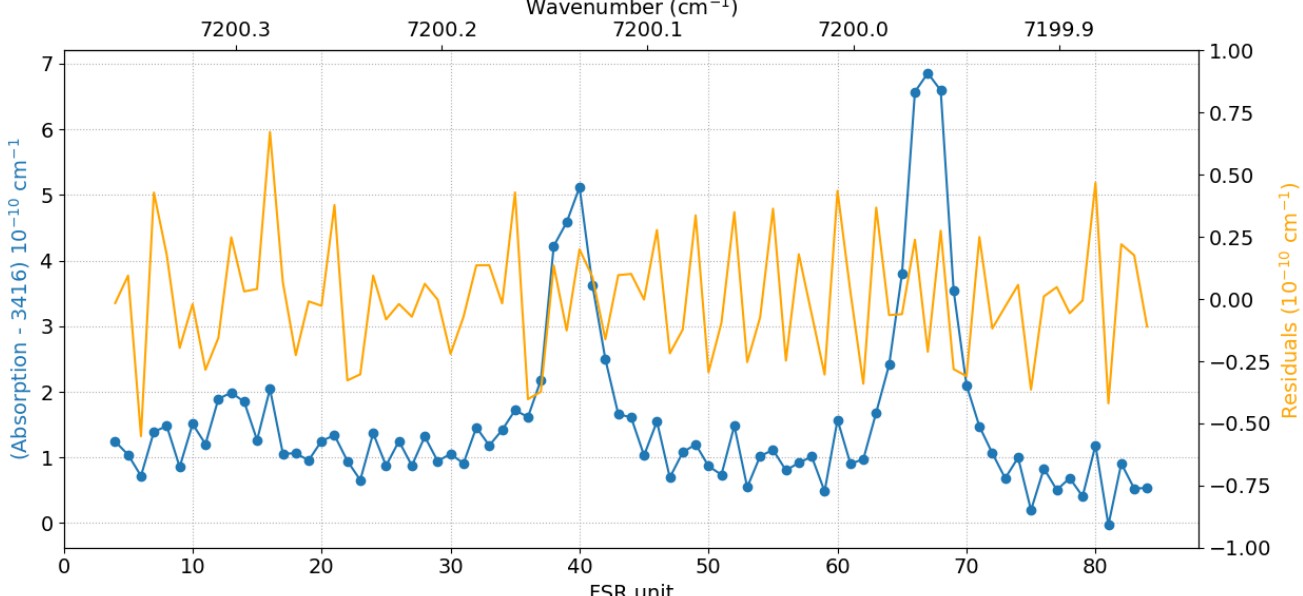

**Figure 2 : Measured spectrum (after correction by a background absorption offset of 3 416 x 10⁻¹⁰ cm⁻¹) of the OF-CEAS analyser (blue dots) after a long drying period using a N₂ gas cylinder, resulting in a minimal water concentration of 3 ppm. The residuals after fitting are expressed in cm⁻¹ (yellow line) and obtained from a 600 ms wavelength scan and a fit calculation time of less than 52 ms.**

For a ring-down of 98 µs, and an acquisition of 80 modes, the wavelength scan is performed within 600 ms to enable at the same time a useful signal to noise ratio and an interesting time resolution for the analysis of transient water vapour phenomena. In order to keep the data acquisition fast and in real time, the fitting algorithm is tuned by fixing most parameters. The typical calculation time is 52 ms in steady operation, which is shorter than the wavelength scan time.

## 1.2 Low humidity level generator

For continuous water vapour isotopic measurement, the performance of the analyser must be characterized in terms of stability over time (Allan deviation; Werle et al., 1993) and water mixing ratio (hereafter called humidity) dependency of the isotopic measurements (Weng et al., 2020). Additionally, during in-situ measurements, a periodic calibration at one specific humidity level is required for drift correction, as the optical signal can be affected by several time-dependent factors, such as temperature or mechanical perturbations.

The characterization of the instrument is performed with a custom laboratory *low humidity level generator* (LHLG) (Leroy-Dos Santos et al., 2021), which enables the generation of a steady water vapour flux with a known and stable isotopic value. A water droplet is generated at the tip of a needle inside an evaporation chamber, flushed by a controlled dry air flux. By controlling both the water and air fluxes, it is possible to precisely control the humidity content of the generated moist air, while the isotopic value is defined by the water sample (Kerstel, 2021) .

The calibration results shown in this paper are carried out with a new version of the LHLG. An updated architecture gives easy access to the various elements of the instrument (including electronics), while remaining compact and adapted for field operation. Among its new features, the evaporation chambers are now equipped with cartridge heaters to reach higher humidity levels. With a regulated temperature of 60°C inside the evaporation chambers, a stable humidity above 10 000 ppm can be reached, whereas the older version was limited to a maximum humidity of ~ 2 000 ppm. A sequencer was also implemented in the LHLG software, enabling long calibrations with automatic syringe refill cycles. This allows the assessment of the spectrometer stability on longer timescales, with several days of stable standard injection.

## 2 Performance of the instruments

In this section, we present the characterisation results of two OF-CEAS instruments manufactured by the AP2E company. The first analyser, which we will refer to as "high-humidity analyser" (serial number #1087) has a cavity ring-down time of 54 µs
(cavity finesse of 64 000) and was installed in December 2022 at the Dumont d'Urville station (66°40′ S, 140°01′ E) and characterised during the austral summer seasons 2022-2023 and 2023-2024. The second analyser, featuring higher reflectivity mirrors, has a ring-down time of 98 µs (cavity finesse of 116 000) and was entirely characterised in the laboratory. This analyser will be referred to as "low-humidity analyser" (serial number #1169).

### 2.1 Time stability

To quantitively assess the mid- and long-term stability of the OF-CEAS instruments, we used the LHLG to perform Allan deviation (AD) measurements (from a few hours to one week) and drift measurements over one year with regular automatic calibrations.

### 2.1.1 Allan deviation study

The OF-CEAS stability is assessed at 500 ppm and 100 ppm, which correspond to a LHLG infused water rate of 0.1125 µL/min
and 0.0225 µL/min, respectively. As the LHLG is equipped with 100 µL syringes, a one-week long measurement is performed by generating successive plateaus separated by a gap of ~1-2 hours necessary for the syringe refill and the humidity and isotopic composition stabilisation. Figure 3 shows laboratory measurements with the "low-humidity" analyser of the humidity, the $\delta^{18}O$ and the $\delta D$ at 500 ppm (blue) and 100 ppm (red). For comparison, an additional dataset at 1000 ppm from the "high-humidity" analyser is plotted in green. Very stable plateaus are obtained from the LHLG, reaching a standard deviation of 3.1
ppm at 100 ppm during a one-week sequence.

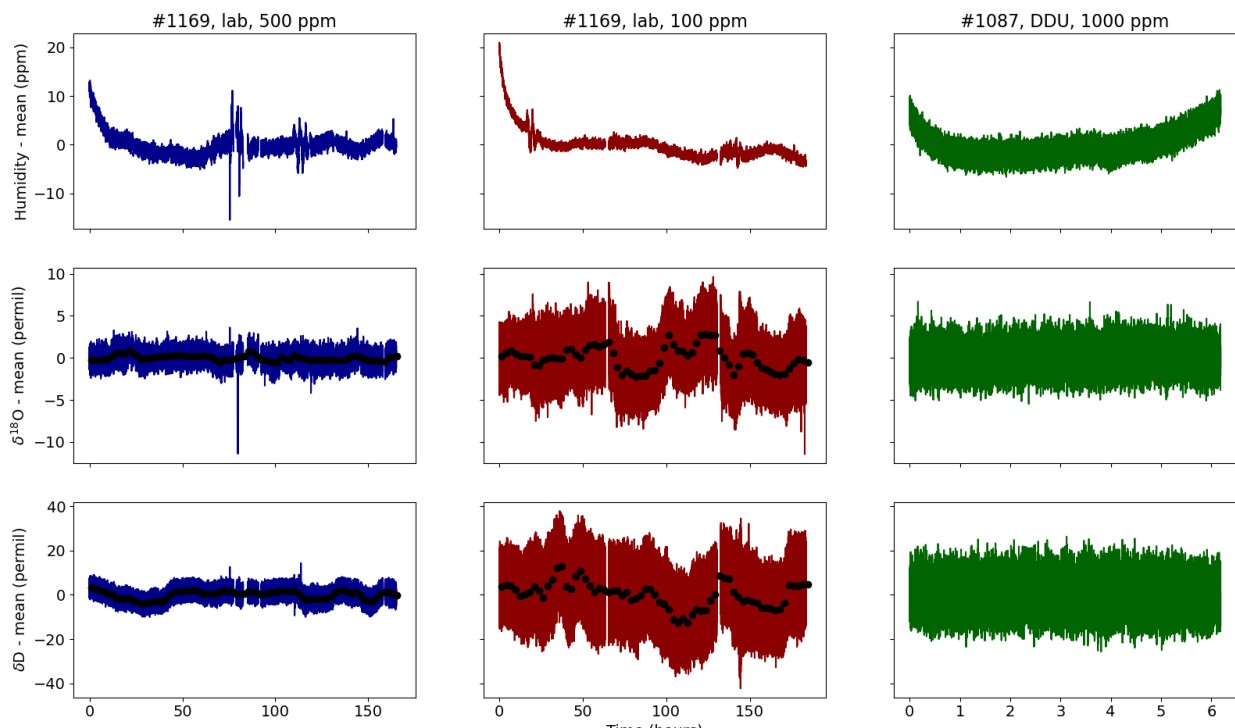

**Figure 3 : From top to bottom: measured humidity, $\delta^{18}$O and $\delta$D used for the Allan deviation study, referenced to their mean value. The first two columns correspond to one-week calibrations in the lab with the low-humidity analyser at 500 ppm (first column, blue) and at 100 ppm (second column, red). The third column (green) corresponds to the data obtained at 1 000 ppm from the high-humidity analyser in the field, over 6 hours. Coloured curves show the raw signal, and the black circles the signal averaged on a 8 000 s window.**

To calculate the long-term Allan deviation of the original data containing gaps (blue and red dataset in Figure 3), a secondary dataset is calculated with a time sampling greater than the gap duration, $\Delta t = 8\,000s$ (black points). The long-term Allan deviation (AD) shown in Figure 4 results from merging the AD of the two datasets. We show in blue and red the long-term AD obtained with the low-humidity analyser at 500 ppm and 100 ppm, respectively. The 500 ppm AD is obtained from a sequence of 14 plateaus with a duration of 13 hours each, while the 100 ppm AD is calculated from 3 plateaus with durations of 65 hours each, and the light colour envelope shows the standard deviation associated with the AD of each individual plateau. Using the second dataset allows for a time range spanning from 8 000 s to two days (empty symbols in Figure 4). For comparison, we added in green the AD obtained from a 6-hour sequence performed at 1000 ppm with the high-humidity instrument (ref. #1087).

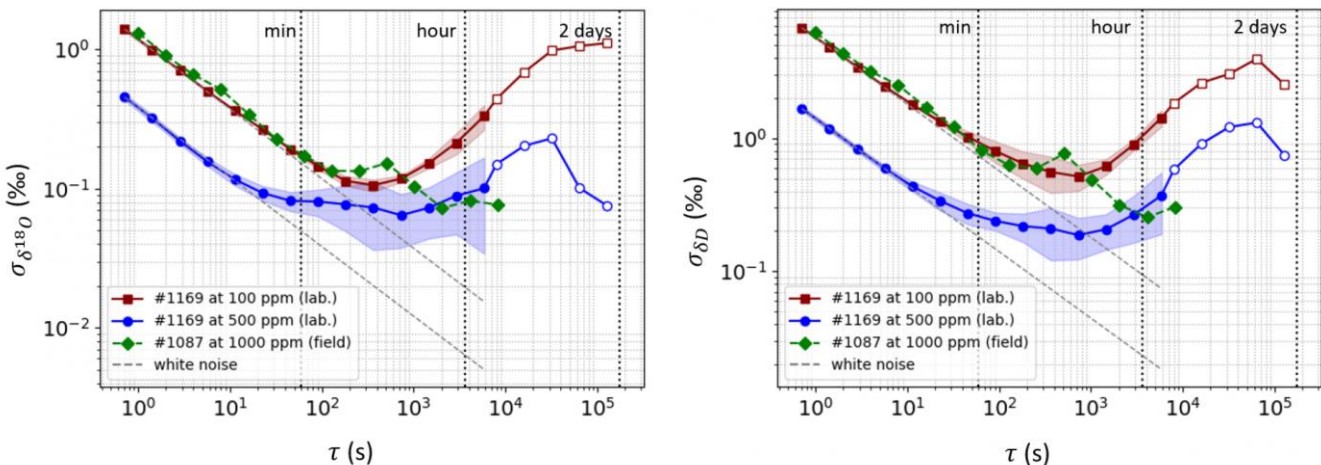

Figure 4 : Allan deviation of δ¹⁸O (left) and δD (right) for a 1000 ppm step performed in the field with the high-humidity analyser (green diamonds, ref #1087) and a 500 ppm sequence (blue circles) and 100 ppm sequence (red squares) performed in the laboratory with the low-humidity analyser (ref. #1169). The empty symbols correspond to the long-term AD performed with a sampling time of 8000s. The grey dashed lines indicate the white noise law $1/\sqrt{\tau}$. The data retrieved from the high-humidity analyser was archived with a sampling time of 1 s, and of 0.7 s for the low-humidity analyser.

The ADs of the low-humidity analyser follow a white noise decay during several minutes, with a minimal value for δ¹⁸O of 0.1 ‰ at 100 ppm and 0.06 ‰ at 500 ppm (0.5 ‰ and 0.2 ‰ for δD at 100 ppm and 500 ppm, respectively). A drift is observed after approximately 10 min, which we attribute to parasitic interferences arising along the optical path between laser and cavity. After a few hours, we observe that the interference phenomena average out, leading to a reduction of the drift slope over long timespans. At a delay of two days, we observe an AD for δ¹⁸O of 1 ‰ at 100 ppm and 0.09 ‰ at 500 ppm (2.5 ‰ and 0.7 ‰ for δD at 100 ppm, and 500 ppm, respectively). For comparison, the maximum values of AD for δ¹⁸O between $10^4$ and $10^5$ s are 1 ‰ (100 ppm) and 0.23 ‰ (500 ppm); for δD, 3.9 ‰ (100 ppm) and 1.3 ‰ (500 ppm). Finally, the AD of the high-humidity instrument (54 µs ring-down) at 1000 ppm shows a white noise equivalent to the 100 ppm AD obtained from the low-humidity instrument (98 µs ring-down). We also note that no particular drift is observed on the high-humidity instrument on the time scale of a few hours because 6 hours is too short to observe mid-term perturbations. This comparison shows that increasing the cavity ring-down time leads to an increase of the signal–to–noise ratio and confirms thus the need for high reflectivity mirrors to target high sensitivities in low-humidity environments.

### 2.1.2 Long-term stability at Dumont d'Urville station

During in-situ measurements, a periodic calibration is performed to check and correct if necessary for instrumental drift on longer time scales, caused by internal instabilities originating from the instrument like parasitic interferences or external perturbations (lab temperature, vibrations, etc). Since this paper presents the first field deployment of an OF-CEAS instrument dedicated to $H_2O$ isotopic analysis, long-term drift was a particular concern, requiring a quantitative study. At the Dumont

d'Urville (DDU) station, the periodic drift calibration consists in a first step of drying (45 minutes) to remove residual atmospheric water vapour isotopes, and two successive steps with two standards injected at a humidity of 1000 ppm (110 minutes in total). This calibration sequence has been set every 46 hours and is the result of a compromise between frequent calibrations and the time dedicated to atmospheric data acquisition.

|  | δ¹⁸O | δD |
|---|---|---|
| Ross 7 | (-18.94 ± 0.05) ‰ | (-146.0 ± 0.7) ‰ |
| AO1 | (-30.60 ± 0.05) ‰ | (-238.3 ± 0.7) ‰ |
| TD3 | (-40.19 ± 0.05) ‰ | (-313.6 ± 0.7) ‰ |
| FP5 | (-50.52 ± 0.05) ‰ | (-394.7 ± 0.7) ‰ |
| OC4 | (-53.93 ± 0.05) ‰ | (-422.7 ± 0.7) ‰ |

**Table 1: List of in-house standards used in this study and their VSMOW/SLAP calibrated δ¹⁸O and δD values (determined with a Picarro 2130-i analyser for δD and a Finnigan MAT252 mass spectrometer for δ¹⁸O).**

In Figure 5, we present the calibration points performed over the year 2023 – with a gap from mid-June to mid-July due to a breakdown of the LHLG – using two in-house standards (Table 1) calibrated against the VSMOW/SLAP scale, FP5 ($\delta^{18}O$ = - 50.52 ± 0.05 ‰ and $\delta D$ = - 394.7 ± 0.7 ‰), and AO1 ($\delta^{18}O$ = - 30.6 ± 0.05 ‰ and $\delta D$ = - 238.3 ± 0.7 ‰). The OF-CEAS calibrations are compared to the values obtained with a L2130-i Picarro instrument (CRDS technology) already running in this station (Leroy-Dos Santos et al., 2023). The generated humidity values across the one-year isotopic calibrations have a good repeatability, with a typical standard deviation of 30 ppm around the theoretical setpoint of 1000 ppm. After filtering to remove the calibrations with a non-stable humidity (i.e. standing outside the 2-$\sigma$ interval), we obtain 138 calibrations for the OF-CEAS analyser and 146 calibrations for the CRDS analyser. Each point in Figure 5 corresponds to the $\delta^{18}O$ (top panels) and $\delta D$ (bottom panels) mean values taken over a 5 to 10-minute window at the end of the humidity step, with blue circles for the OF-CEAS analyser (AP2E company) and green circles for the CRDS analyser (Picarro company).

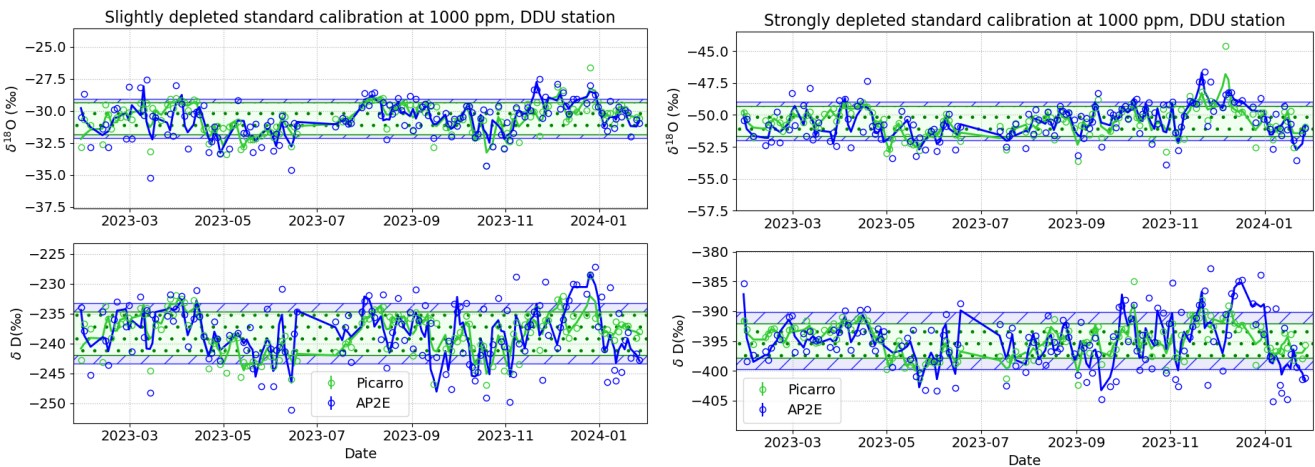

**Figure 5 : $\delta^{18}O$ and $\delta D$ drift calibration of the AP2E OF-CEAS (blue circles) and Picarro CRDS (green circles) analysers for two in-house isotopic standards named AO1 (slightly depleted standard) and FP5 (strongly depleted standard). Each calibration point represents the average of the final 5 to 10 minutes of humidity plateaus, with a humidity setpoint of 1000 ppm. The blue line corresponds to the AP2E OF-CEAS dataset smoothed over a 5-point window, and similarly the green line corresponds to the smoothed Picarro CRDS dataset. The blue hatched (resp. green dotted) area corresponds to the standard deviation of the AP2E OF-CEAS (resp. Picarro CRDS) calibrations, with the corresponding values displayed in Table 22.**

The resulting data show no long-term trend on a one-year range for either the AP2E or the Picarro instrument, with a higher dispersion of the OF-CEAS dataset (Table 2). We note also that the analysers' calibrations show a correlation on a monthly

scale, which could indicate a drift of the calibration instrument. Indeed, large temperature variations have been registered inside the shelter (5°C of maximal amplitude during summer season), which has an impact on the time response of the calibration plateaus and thus the value of the isotopic composition at the end of the plateau. This underscores the need for a temperature regulation in the building housing the instruments at DDU and/or inside the evaporation chamber of the calibration instrument.

| | OF-CEAS (AP2E) – 138 calibration points | | CRDS (Picarro) – 146 calibration points | |
|---|---|---|---|---|
| | $\sigma(\delta^{18}O)$ | $\sigma(\delta D)$ | $\sigma(\delta^{18}O)$ | $\sigma(\delta D)$ |
| Slightly depleted standard (AO1) | 1.6 ‰ | 4.9 ‰ | 1.2 ‰ | 3.7 ‰ |
| Strongly depleted standard (FP5) | 1.5 ‰ | 4.5 ‰ | 1.2 ‰ | 3.0 ‰ |

**Table 2: Standard deviation of the two standards isotopic calibrations shown in Figure 5 performed from January 2023 to January 2024 on the OF-CEAS and CRDS analysers.**

**2.2 Humidity and isotopic composition dependency**

In this section, we present the characterisation referred to in the literature as the mixing ratio dependency, which is used in various atmospheric isotopic measurements such as those of $O_2$ (Piel et al., 2024), $CO_2$ (Flores et al., 2017) or $H_2O$ (Weng et al., 2020). Indeed, for a water vapour sample with a given isotopic composition, the measured isotopic ratio can be affected by the humidity level (through different processes, such as spectroscopic effect affecting the fitting procedure or memory effect). In addition, this humidity dependency can differ for different isotopic ranges, especially at low humidity content (Casado et al., 2016; Leroy-Dos Santos et al., 2021; Weng et al., 2020). We use in this study the most common, so-called "ratio method", which consists in calculating first isotopic ratios from the measured optical spectrum, and then correcting them from the mixing ratio dependency. We determined the humidity dependency calibration with two water isotopic standards corresponding to the expected isotopic range in the field. Two in-house standards (calibrated against the VSMOW/SLAP scale, Table 1) are used for the laboratory calibration: the OC4 standard, strongly depleted and adapted for measurement on the Antarctic plateau ($\delta^{18}O$ = -53.93 ± 0.05 ‰, $\delta D$ = -422.7 ± 0.7 ‰) and a slightly depleted standard, ROSS7, close to the water vapour isotopic composition of coastal Antarctic sites ($\delta^{18}O$ = -18.94 ± 0.05 ‰, $\delta D$ = -146.0 ± 0.7 ‰). Field calibrations are also presented in this section, using the additional calibrated standards AO1 and FP5 covering a similar range (Table 1).

First, the humidity dependency of $\delta^{18}O$ and $\delta D$ is established taking as a reference the measured value at a given humidity $h_{ref}$, generally chosen in the range of observed values at the site of interest (Figure 6). A fit of the calibration points gives the correction function $f_{calib}$, which verifies the condition $f_{calib}(h_{ref}) = 0$ and is further used for correcting the acquired isotopic data. As the humidity dependency can be different from one standard to another, different strategies can be used to estimate the correction function (Weng et al., 2020). If the expected isotopic range is narrow enough or the correction functions are similar from one standard to another, a global fit using the data of several standards in an undifferentiated way can be performed. In the case of divergent correction functions, it is more reliable to make a humidity dependency calibration with two standards, and then define a general, two-dimensional calibration function, defined as the linear interpolation between the two correction functions. Once the calibration points are fitted, for a given humidity $h$ and measured isotopic value $\delta_{raw}$, the data is corrected as follows:

$$\delta_{corr} = \delta_{raw} - f_{calib}(h, \delta_{raw})$$

In Figure 6 we show the characterisation obtained with the analyser adapted for low humidity (left column) and for high humidity environments (right column). The calibration of the low-humidity analyser was performed in the lab, and repeated

several times within a 3-month period, with the very depleted standard OC4 (blue circles) and the slightly depleted standard ROSS7 (red squares) and a reference humidity fixed at 500 ppm. For the low humidity analyser, shades of blue and red indicate the various measurements acquired between March (light colour) and May 2023 (dark colour). To reduce potential sample-to-sample effects that are more likely to arise below 500 ppm, the calibration always starts with the high humidity steps (above 1000 ppm) and finishes with the low humidity step (50 ppm), meaning that the tubings have been flushed with the same standard for at least 10 hours before the last calibration point. With this set-up, we observe the same humidity dependency trend across three months, even when alternating the order of the standard injection, which confirms the absence of sample-to-sample effect. The high-humidity analyser calibration was performed at Dumont d'Urville station during a 48-hour-long sequence, using the strongly depleted standard FP5 (blue triangles) and the slightly depleted standard AO1 (red diamonds), in December 2022 (light colour) and December 2023 (dark colour). An initial humidity sequence was performed in the low humidity region (50-1 500 ppm) and a second run for the high humidity region (2 000 – 6 000 ppm), using a heated evaporation chamber (60°C). The vertical red dotted line indicates that more than 99% of the absolute humidity values measured over the year at DDU stay above this threshold, i.e. in the linear region. A reference humidity of 1 000 ppm was chosen here, as it lies closer to the average humidity values measured on the Antarctic coast.

Two distinct regimes can be highlighted from the humidity dependency (Figure 6). Below 500 ppm, we observe a divergence of the isotopic value (here assimilated to a 1/x function) with distinct trends for the strongly depleted (blue) and slightly depleted (red) standards. Above 500 ppm, the two curves merge and a linear dependency for $\delta^{18}O$ and $\delta D$ is observed on both instruments. For the high-humidity analyser, we observe a good superposition for the humidity response of the two standards AO1 and FP5 from 6 000 to 500 ppm, corresponding to more than 99% of the humidity values usually recorded at DDU station. The measured slopes of the humidity response in the 500 – 6 000 ppm region are reported in Table 3.

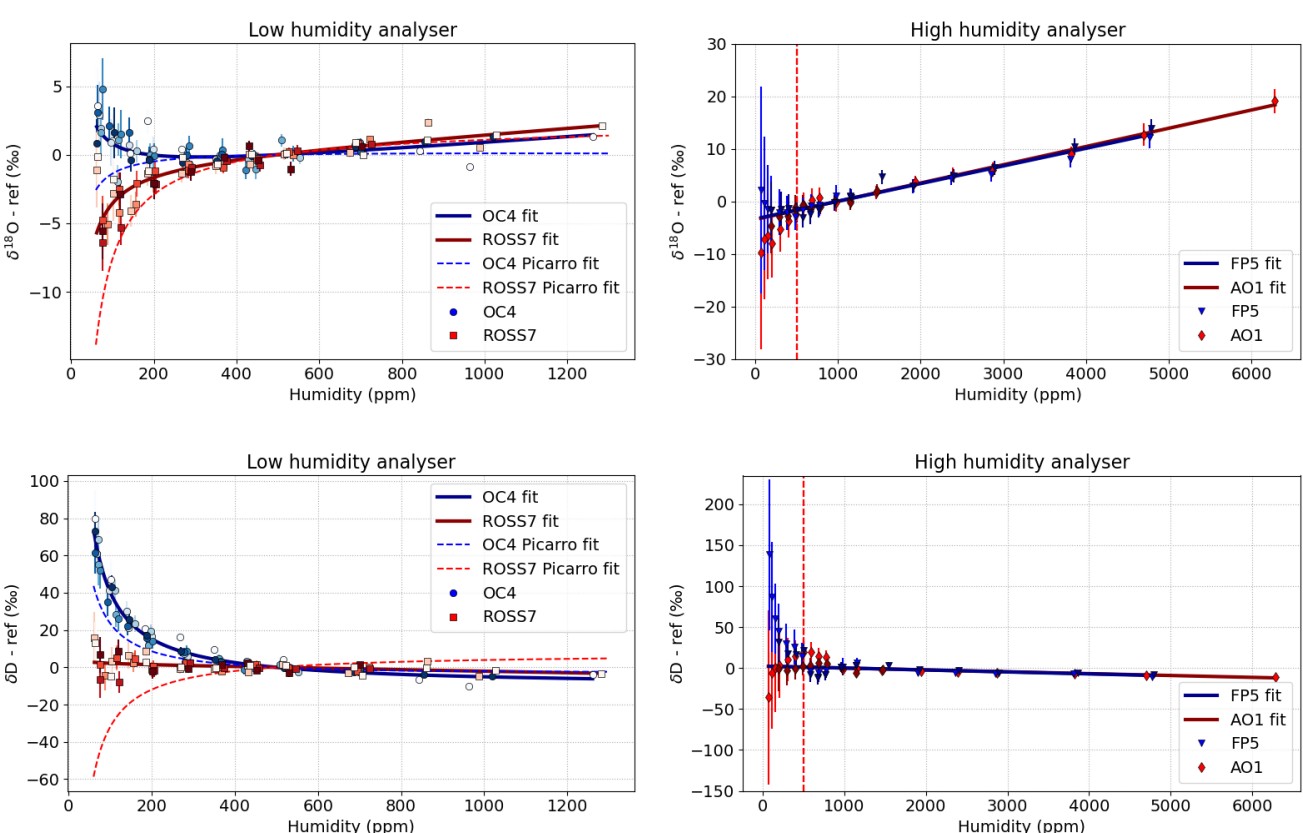

**Figure 6 : Humidity dependency calibration of the low-humidity OF-CEAS analyser from 50 to 1 300 ppm for $\delta^{18}O$ (top left) and $\delta D$ (bottom left) and of the high-humidity analyser from 50 to 6 500 ppm for $\delta^{18}O$ (top right) and $\delta D$ (bottom right). The y-axes are shown with different scales. All curves are referenced to the isotopic composition measured at 500 ppm (left panels) and 1000 ppm (right panels), denoted "ref" on the y-axis. Shades of blue and red indicate various calibration sequences in time. Additional dashed**

 **curves correspond to the typical humidity dependency of a Picarro instrument measured in the lab, for comparison. The vertical dashed line in the right panels corresponds to the humidity above which 99% of the humidity signal at DDU is observed**.

The positive slope on the $\delta^{18}O$ calibration curve is explained by the presence of a strong absorption line of water located around 1389 nm (as shown in Figure 1), creating a shift of the baseline and a bias on the fit, while for δD this creates a negative slope. As the HDO absorption line is situated further away from the large water absorption peak, the slope has a smaller amplitude. Below 500 ppm, we observed a larger noise on the high-humidity analyser (#1087) installed at DDU, which features lower reflectivity mirrors (ring-down of 54 µs) than the low-humidity analyser (#1169, ring-down of 98 µs) characterized in the laboratory.

| | Slope from high-humidity #1087 analyser (‰ / 1000 ppm) | | |
|---|---|---|---|
| | **Slightly depleted standard (AO1)** | **Strongly depleted standard (FP5)** | **Mean** |
| $\delta^{18}O$ | 3.1 | 3.0 | 3.1 |
| $\delta D$ | -2.7 | -2.2 | -2.5 |

**Table 3: Slope of the humidity dependency calibration for the high humidity spectrometer, in the 500 – 6 000 ppm region, expressed in ‰/1 000 ppm.**

The characterisation performed on both analysers highlights that, over a one-year timespan, no significant drift is observed between the humidity dependency calibrations, and that a global linear correction function can be applied above 500 ppm. Below 500 ppm, we need to consider the divergence between the two standards by using a two-dimensional correction function defined as the linear interpolation between the slightly depleted and the strongly depleted standard correction function.

**2.3 Instrument accuracy against the VSMOW/SLAP scale**

We demonstrated the stability of the instrument for short to mid-term time spans with the Allan deviation and for longer time periods with repeated humidity calibrations during one year. After having estimated the humidity dependency correction of the OF-CEAS analyser, we present in this section the instrument accuracy against the VSMOW/SLAP scale, using a linear calibration from two standards, following the NIST recommendation (Reference Material 8535). An additional standard situated within the isotopic range is used to quantify the precision and accuracy of the measure.

Figure 7 shows the relation between the measured isotopic value and the true value for the two standards OC4 and ROSS7, and the measurements of an additional standard TD3 for various humidity steps ranging in the divergence area, from 67 to 698 ppm (isotopic composition of the standards in Table 1). From the linear relationship obtained with OC4 and ROSS7 (black dashed line), the expected value for TD3 (red triangle) shows an accuracy of -0.7 ‰ for $\delta^{18}$O and 1.7 ‰ for δD, compared to the independent VSMOW/SLAP calibrated value, $\delta^{18}$O = - 40.19 ± 0.05 ‰ and δD = - 313.6 ± 0.7 ‰, and a precision in this humidity range of 0.4 ‰ for $\delta^{18}$O and 3.6 ‰ for δD.

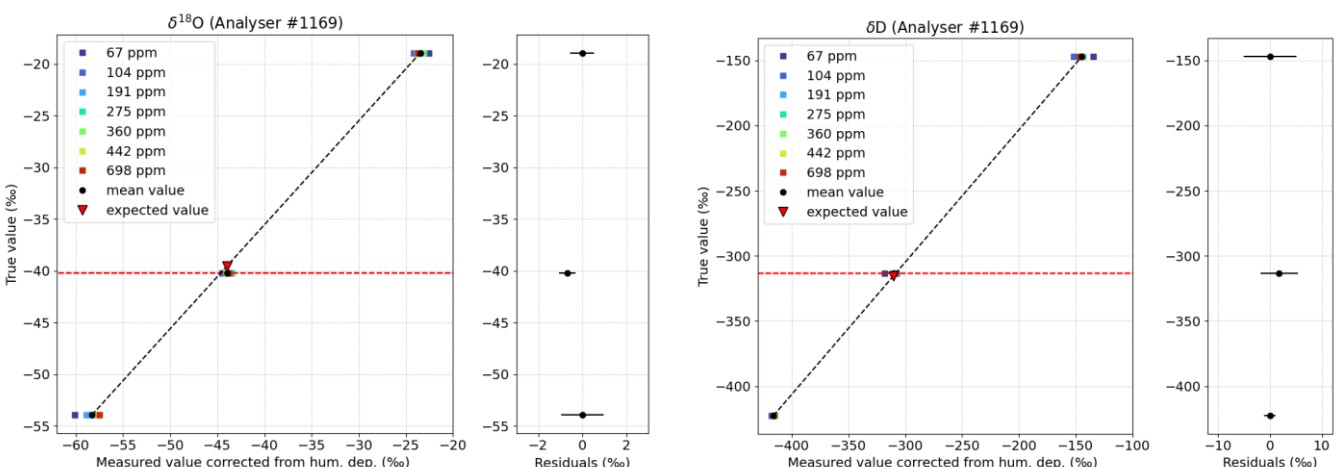

**Figure 7: Correspondence between the true isotopic value and the measured, corrected value of $\delta^{18}O$ and $\delta D$ for 7 humidity steps ranging from 67 to 698 ppm with three VSMOW/SLAP calibrated standards OC4, TD3 and ROSS7 (coloured squares), with the corresponding average values calculated across the humidity range (black circles). The linear calibration slope (black dashed line) results from the average value of the OC4 and ROSS7 standards only, while the TD3 standard (true value indicated with the red dashed line) is used to quantify the instrument accuracy and precision. The red triangle indicates the expected value of the TD3 standard using the calibration slope.**

## 3 Discussion

### 3.1 Expected performance for in situ water vapour isotope measurement in the frame of the AWACA project using OF-CEAS technology

In addition to the already installed analyser at DDU station, several OF-CEAS analysers will be deployed during the austral summer 2024-2025 in remote sites, from the Antarctic coast (DDU station) to the plateau above 3200 m (Concordia station). The three chosen remote sites, named D17, D47 and D85, as well as Concordia station (DC) are shown in Figure 8. The instrumental deployment will be achieved in the framework of the ERC (European Research Council) AWACA (Atmospheric Water cycle over Antarctica) project. This project aims to advance the understanding of the dynamical and physical processes affecting the quantity, phase and isotopic composition of water along the atmospheric branch of the Antarctic water cycle, including snow-atmosphere exchanges, from the coast to the inland plateau. For this purpose, isotopic measurements will be integrated with other atmospheric measurements (surface meteorology, cloud and precipitation properties) and the new datasets will be used to improve the related parameterizations of state-of-the-art regional and global atmospheric models.

To give a quantitative overview of the expected performances for this deployment, we calculated for each site the proportion of days per year with average humidity below 500, 100 and 10 ppm (retrieved from automatic weather stations in 2018 for D85 and in 2020 for the other sites; see Figure 8, left). For each humidity value, we estimated the standard deviation after 24 hours of integration from the long-term AD measurement performed on the low-humidity analyser using the method presented in part 2.1.1. At 500 ppm and 100 ppm, the LHLG enables repeated injections of ROSS7 standard. An additional step at 10 ppm is performed and corresponds to residual water obtained by a pure drying using the LHLG without any water sample injection. We plotted in Figure 8 (left) the estimated standard deviation of $\delta^{18}$O (resp. δD) in red (resp. dark red).

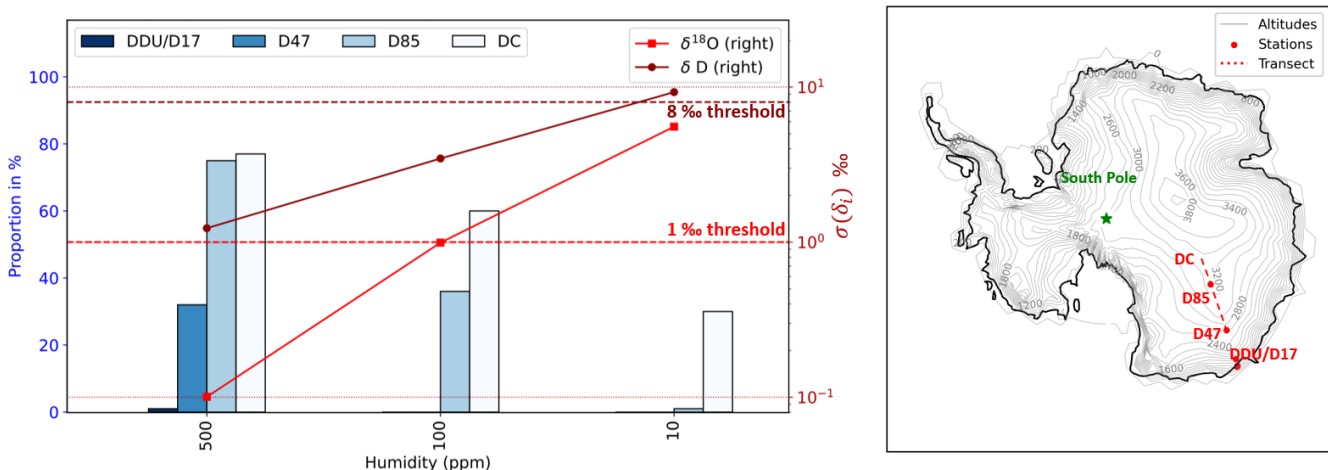

330

**Figure 8 : On the left, histogram representing the year fraction (expressed in %) below a fixed humidity content for 4 sites situated along the transect. For each humidity, we plotted the associated standard deviation after 24 hours σ(δ$_i$) as predicted by the Allan deviation study for δ$^{18}$O and δD. The dashed horizontal lines represent the δ$^{18}$O and δD upper thresholds for the standard deviation to confidently study the diurnal cycle. These thresholds are set at a value 10 times lower than the amplitude of the diurnal cycle, to ensure a proper signal resolution (see discussion in the text). On the right, map with the location of the 4 instrumented sites for the AWACA deployment.**

As typical diurnal cycles of the water vapour isotopes are of order 10 ‰ for $\delta^{18}$O (resp. ~ 80 ‰ for $\delta$D) at Dumont d'Urville and Dome C – Concordia (Bréant et al., 2019; Casado et al., 2016), we suggest a noise threshold of 1 ‰ for $\delta^{18}$O and of 8 ‰ for $\delta$D, above which we consider that no interpretation of the isotopic signal at the diurnal scale can be confidently made.

These threshold values are indicated in the figure by the horizontal dashed lines. We observe on average a larger noise for $\delta$D, explained by the smaller absorption intensity of the HDO line compared to the $H_2{}^{18}$O line. However, while the $\delta^{18}$O deviation crosses the threshold noise at around 100 ppm, the $\delta$D deviation stays below the 8 ‰ threshold, until approximately 10 ppm. We can conclude from this characterization that we should prefer acquisitions of $\delta$D over measurements of $\delta^{18}$O in very dry environments.

The above characterisation leads us to propose the following calibration scheme for water vapour isotope monitoring in Antarctica:

- The humidity dependency shows no particular drift on a 1-year period, so we suggest a humidity-isotope dependency calibration every year using two standards, in the humidity and isotopic range of the site of interest.

- The drift calibration should be performed preferably every 24 to 48 hours, to correct for mid-term drift while keeping enough time for data acquisition.

With this calibration scheme and using the noise estimation from the Allan deviation study as a criterion to study diurnal cycles, we expect enough resolution on the isotopic signal down to humidity values around 10 ppm for $\delta$D and 100 ppm for

$\delta^{18}$O. This estimated limit of detection opens up the possibility of studying the cycle of water isotopes in Antarctica all year round from the coast to D85 station, and about 70% of the time at Concordia station. We would like to point out that this limit of detection considers the intrinsic limit of the OF-CEAS instrument, but does not include the low humidity calibration uncertainty (e.g., gas matrix effect, residual water mixing), which will be discussed in the section below.

### 3.2 OF-CEAS performance and comparison with commercial CRDS technique

**Signal stability and noise**

On short time scales, the OF-CEAS technique allows for high isotope-ratio precision at low water concentrations. In Figure 9, we compare the Allan deviation value at 2 minutes integration of the commercial CRDS instrument (Picarro) installed at DDU station, and of the two OF-CEAS instruments (AP2E). From 60 to 3000 ppm, the low-humidity OF-CEAS analyser equipped with high reflectivity mirrors shows a noise reduction by a factor of approximately 5 compared to the CRDS and the high-

365 humidity OF-CEAS analyser. This shows the ability for the OF-CEAS technique to capture transient events at high precision, and demonstrates the potential of the instrument, in particular in the low humidity range where the noise increases exponentially.

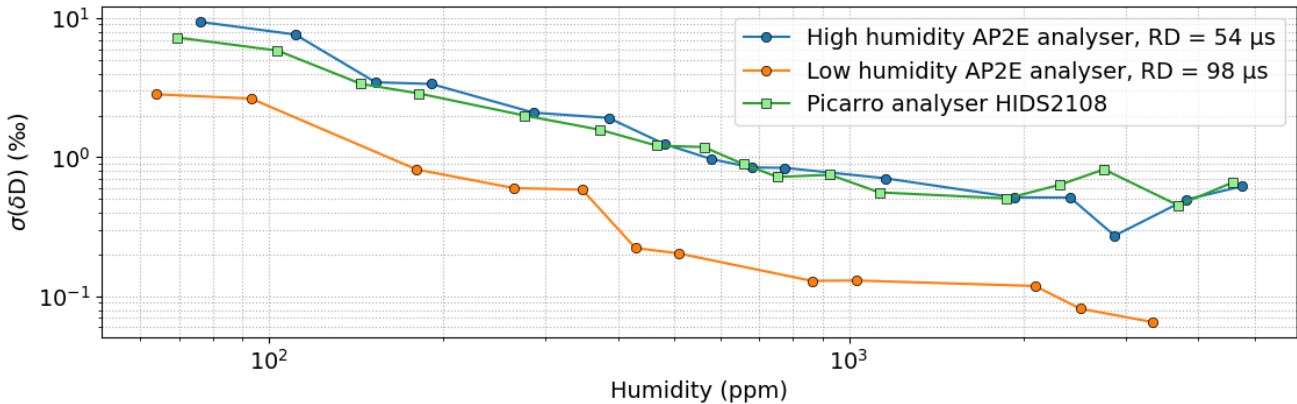

**Figure 9 : Noise on $\delta D$ with an averaging time of 2 min as a function of the humidity for the two OF-CEAS analysers from AP2E and the Picarro analyser currently installed at Dumont d'Urville station. The noise is obtained from short-term Allan deviations at $\tau$ = 2 min calculated during each step of the humidity dependency calibration.**

On longer timescales, the calibration performed at DDU during 1 year shows no long-term drift on both isotopologues (Figure 5) neither for the commercial OF-CEAS nor the CRDS instrument, although we observe a higher dispersion on the OF-CEAS dataset (Table 2).

The particular feature of the OF-CEAS technique is well illustrated by the Allan deviation study presented in section 2.1.1. It shows an optimum stability range of ~ 15 min, followed by a drift period in the hour range and finally a reduction of the drift in the day range. We showed in the section above that a calibration every 24 hours, resulting from a compromise between precision and data acquisition time, allows for enough precision to interpret more than 70% of the yearly data on the East Antarctic plateau. However, to get the most out of the analyser precision, a calibration would have to be performed every 10 – 20 minutes, which is not compatible with continuous water isotope measurement.

Unlike the CRDS technique which is based on ring-down time measurements to quantify the water isotope concentration, the OF-CEAS technique directly measures the transmitted light intensity. This leads to a very fast response and a low instantaneous noise signal, at the expense of a higher sensitivity to interferences. Indeed, the noise measured in OF-CEAS instruments originates from instabilities encountered in the hourly range, as highlighted in the AD determination, and which we attribute essentially to parasitic interferences. Parasitic interferences in OF-CEAS instruments come from reflective surfaces situated between the laser and the photodetector, like mirror mounts, polarizers or the metallic gas cell, and can affect the signal. Such interferences are sensitive to temperature variations that can occur especially along the laser to optical cavity path. Two main levers have been identified to optimize the precision of the measurements with the OF-CEAS analyser:

- Increase the optical signal stability, by reducing the interferences (efficient optical absorption and thermal regulation, use of low thermal expansion materials), or correct them (use of a reference photodetector, signal post-processing)
- Increase the calibration frequency, by optimising the LHLG settings and reducing the calibration time. For example, using a one standard calibration instead of two, and sending dry air through the humidity generator chambers and tubing before sending it to the instrument could lead to a calibration time reduction from 2:35 hours to 40 minutes, which could be performed every 10 hours while maintaining a good time resolution. To further reduce the uncertainties of the calibration and generate identical humidity calibration plateaus over the whole year, a temperature regulation of the evaporation chamber of the LHLG is also preferred.

**Humidity dependency and calibration uncertainty**

The characterization of the two analysers showed a linear humidity dependency for humidity levels above 500 ppm. Below 500 ppm, the humidity dependency diverges for different isotope ratios. The divergence at low humidity is also observed on commercial CRDS analysers as shown in Figure 6, in particular because both techniques can be affected by a biased fit. Indeed,

it has been shown (Johnson and Rella, 2017; Weng et al., 2020) that broadening or narrowing of the absorption lines and baseline shift due to a changing gas mixture can affect the fitting, thus inducing an error and leading to a humidity and isotope dependency of the measured isotopic composition.

However, to the best of our knowledge, such studies were limited to a minimal humidity value of 500 ppm in most of the calibrations (and 300 ppm for only one calibration; Weng et al., 2020). In the case of humidity values below 300 ppm, we think that residual water with a fractionated isotopic composition mainly driven by ambient air in the injection set-up can be mixed with the calibration standard and thus affect the measured signal by shifting upwards the most depleted standard isotopic ratio and downwards the less depleted isotopic ratio (see Figure 6). This is often mentioned as memory effect in the literature (Bailey et al., 2015). A pure drying using the LHLG in the laboratory (with a typical ambient water mixing ratio of 15 000 ppm) without any water injection leads to a residual water of 10 ppm. We can suppose that in this case, the isotopic composition for humidity levels below 100 ppm is affected in a non-negligible way by the memory effect, i.e. results from a mixing between the injected standard and residual water.

This low humidity mixing effect adds calibration uncertainty, although it is expected to be limited in the field because of a lesser difference between the water mixing ratio inside and outside the instrument. Another source of uncertainty in the low humidity region is the gas matrix, and in particular a possible methane contribution (see Figure 1) affecting the spectrum baseline or the water absorption lines width and thus the isotopic ratios. Finally, slight misalignment of the optical components of the instrument after transport (caused by vibrations or thermal expansion) can also impact the transmitted optical spectrum and thus affect the humidity dependency. We emphasize the importance of calibrating the instrument in the field to best correct for these artefacts (Casado et al., 2016). Such artefacts are difficult to evaluate and a dedicated study is still missing to quantify the resulting calibration uncertainty, which increases when humidity values drop below 100 ppm. We think that future studies focused on the low humidity residual water mixing effect, as well as the impact of methane, would improve the accuracy of water vapour isotopic records measured in extremely dry environments.

**Interesting features for field operation**

For field deployment in extremely dry conditions, and for the particular application of water vapour isotope measurement in Antarctica, we found a great interest in using AP2E OF – CEAS analysers instead of Picarro CRDS. AP2E spectrometers are made up of optical parts that are mainly assembled mechanically, rather than glued together as is the case in Picarro spectrometers, making it possible to perform fine optical adjustments in the field or even to remove mirrors for cleaning in case of contamination. The internal architecture of these analysers therefore reduces the risk of breakdowns during the deployment, but requires an expertise to finely tune them. The embedded software offers the possibility to tune several regulation parameters like various temperature and pressure set points and PID parameters to adapt the analyser to the local conditions in the field. Finally, a large number of internal variables are accessible on the instrument, making it possible to quickly diagnose the state of the instrument in the event of a breakdown in the field.

**Conclusion**

This paper is focused on the characterisation and performance of two water vapour isotope analysers based on new commercial laser spectrometers using the OF-CEAS technique, particularly adapted for dry regions. The first, "low-humidity" analyser featuring high reflectivity mirrors, has been fully characterised in our laboratory: it shows a low limit of detection and is thus specially adapted to very dry regions such as the East Antarctic plateau. The second, "high-humidity" analyser with slightly lower reflectivity mirrors was installed during the austral summer 2022-2023 in Dumont d'Urville, a coastal station of Antarctica, where humidities are not so low.

The stability of the OF-CEAS analysers has been detailed through a long-term Allan deviation analysis using the humidity generator, and an unprecedented one-year long calibration measurement has been performed at DDU station and compared to a commercial CRDS analyser, with no visible long-term drift on either instrument. In addition, the water mixing ratio dependency of the OF-CEAS analysers have been characterized, as well as the accuracy and precision in the low humidity region. We have finally estimated the minimum humidity to confidently interpret diurnal cycles at our sites of interest, namely 100 ppm for $\delta^{18}O$ and 10 ppm for $\delta D$.

Compared to traditional CRDS analysers used for water isotope monitoring, OF-CEAS analysers equipped with high reflectivity mirrors show an extremely low noise, at the expense of a higher sensitivity to any perturbation of their environment like the temperature. This low noise and fast response open up the possibility to measure transient phenomena, like the in-situ measurement of the isotopic composition of individual snowflakes. Moreover, for the particular application of field monitoring in remote areas like Antarctica, these instruments meet the need for an optimizable and adaptable instrument, reducing the risks of breakdown. The OF-CEAS analyser limitations highlighted in this article are the instabilities that develop at the time scale of a quarter-hour or so, which we tentatively attribute to parasitic interferences. Some solutions to reduce these interferences have been identified, such as managing parasitic reflections with optical absorber, improving the thermal stability inside the instrument or installing a reference photodetector. These new developments are currently under study, in order to provide the best possible data for the instrumental deployment of the operational units for the AWACA project, planned for the Antarctic season 2024-2025. Beyond Antarctica, other isotopic water vapour monitoring projects, especially in dry conditions or airborne campaigns, could also benefit from the possibilities of these new instruments.

*Data availability.* The data of this paper is available upon request.

*Author contributions.* TL optimised the OF-CEAS spectrometers, characterised the instruments, produced all the plots, designed and wrote all sections of the original paper, with inputs from co-authors revising the text. TL and EF installed the instrumentation at DDU during the season 2022-2023. MC and TL wrote the code for the Allan deviation calculation. EF and AL made substantial contributions throughout the paper. DR made substantial contributions on section 2.1 and 3. OJ improved the software to control the calibration instrument (HumGen) and developed an additional software for data processing (HumdepApp). FP, OJ and TL fabricated the calibration instrument. GN made the laboratory calibration plotted in section 2.2 and section 2.3. KJ from AP2E designed the ProCeas® analyser. MM and KJ helped for the optimisation of the OF-CEAS analyser. MF and DR designed and optimised the OF-CEAS prototype. VMD (PI of the ERC Synergy Grant AWACA project) and AL designed the AWACA project.

*Competing interests.* This work was made possible by a collaboration with the AP2E company who produced the analysers presented in this manuscript. The characterisation of the analysers was done independently at LSCE, with no interference from AP2E.

*Financial support.* This research was supported by the IPEV ADELISE project, and has received funding from the European Research Council (ERC) under the European Union's Horizon 2020 research and innovation program (AWACA grant agreement No 951596).

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
