# Peer review of "OF-CEAS laser spectroscopy to measure water isotopes in dry environments: example of application in Antarctica"

_EGUsphere, 2024_

## Author Comment (AC1)

This is a well-written, interesting paper about the 'next level' OF-CEAS instrumentation. I liked reading it, and is carries important new information. I recommend publication, but/and have a set of comments and questions that I invite the authors to respond to, preferably by adapting the paper where appropriate, or else in writing to the editor (and me).

We thank Harro Meijer for his very pertinent questions concerning the general form of the manuscript as well as the scientific content, which allow us to be exhaustive on the instrumental limits presented in the paper, and give us interesting perspectives for future studies. The answers are inserted in red and the citations from the manuscript are indicated in blue.

First a general question: how come it is so difficult to perform isotope measurements on water vapour at 10 ppm or even higher, while for example precise isotope measurements on methane (2 ppm) are routine nowadays? Is it because of the humidity range that has to be accomodated?

We thank the reviewer for this remark. What defines the minimal acceptable concentration for isotope ratio determination at a required level of precision, is not only the absolute concentration (expressed in ppm) but also the isotopologue abundance, which is of 1.11 % for $^{13}CH_4$ and of 0.2% for $H_2^{18}O$ and 0.03% for HDO (see Table 1, in The HITRAN2020 molecular spectroscopic database, 2022). This means that at equivalent concentrations, the number of molecules is 5 times lower for $H_2^{18}O$ and 40 times lower for HDO compared to $CH_4$ isotopologues.

Another point to consider could be the intensity of the targeted near-infrared transitions. The spectral area depends on isotopologues availability in a reduced frequency range (limited by the maximal frequency range of the laser source), and is also the result of a number of compromises, such as reducing the intensity of the interfering gasses and reducing spectral congestion. If we take the example of the wavelength range used by Picarro G2201-i, for CH4 isotopes measurement, we found (see in Rella et al. 2015, Fig. 1 and Defratyka et al. 2021) an absorption of ~1E-9/cm at the wavenumber 6029 cm-1 for $^{13}CH_4$, for 2 ppm of concentration. For water, an equivalent absorption would be obtained at a concentration of ~ 100 ppm for HDO and ~ 10 ppm for $H_2^{18}O$.

Finally, we are not familiar with the case of $CH_4$ isotopes, but as mentioned by the reviewer, it is true that working with low mixing ratios with water molecules instead of $CH_4$ adds biases in the calibration because of adsorption on the tubings (resulting in residual water mixing / memory effect, discussed further), which makes it more difficult to obtain a given level of accuracy.

Then in more detail, going through the paper:

Figure 1 The methane interference can potentially be severe for low humidity (1/100 to 1/1000 of the humidity in this plot). Apart from this plot, this interference has not been

discussed or even mentioned anywhere else. Is the methane spectrum always included in, and corrected for in the fits?

The methane spectrum is taken into account in the fits and calibrated with atmospheric air (we added few words on this in subsection 1.1). However, the humidity dependency calibrations performed in the lab do not contain CH4, as synthetic air bottles are used (for more details, see Zero Air and Zero Air Plus here https://www.airproducts.co.uk/gases/zero-air). Using a calibration gas containing a typical atmospheric proportion of 2 ppm of CH4 could bring possible effects at low humidity on the calibration curve. We are investigating how to set this up in the field, since this matrix effect could effectively affect the true value correction. We are developing this point in section 3.2, under the paragraph "Humidity and isotopic composition dependency".

We are confident that this will not change the main conclusion of the article, as considering a potential CH$_4$ effect should not impact the intrinsic linearity and precision of the instrument.

Figure 2 It is not clear to me how the residuals in cm-1 relate to the arbitrary normalized intensity (in arbitrary units). Even the x-axis is in free spectra range steps, not in cm-1. Please clarify.

Thank you for pointing out this inconsistency on the y-axis of the figure. The correct unit is now displayed in cm-1 for the absorption spectrum and for the residuals. We also changed the left side y-axis which is now labeled "Measured absorption".

The x-axis could be expressed in terms of cm-1 or in nm as shown in Figure 1, by knowing the absolute wavelength and that one FSR corresponds to 188 MHz. However, contrary to Figure 1, we indeed do not show the absolute wavelength but rather the relative frequency in units of FSR, as we want to highlight how the spectrum is constructed (i.e. that the sampling points are defined by the absorption value at optical resonances).

Compared to figure 1, there is a flat baseline here in figure 2. How can that be, as all lines in figure 1 will be proportionally lower?

The plotted intensity in Figure 2 corresponds to the measured absorption when pure nitrogen is injected. Fig 2 shows the absorption at 3 ppm H2O, for which the baseline is extremely close to the empty cavity losses corresponding to the visible baseline offset, which is around 0.342E-6/cm expressed in absorption per length unit.

Line 169 and elsewhere: suddenly decimal commas instead of points. Happens more often, with the declaration of the values for reference waters.

This has been corrected.

Figure 5 the number of points for the Picarro instrument are much higher than for the AP2E one. Why? DIfferent strategy?

Table 1 should include the number of measurements (so much more for the Picarro I presume)

Thank you for this comment. The calibration dataset presented in this article covers the period ranging from January 28th 2023 to January 28th 2024. It is composed for Picarro of 161 (AO1 standard) and 158 (FP5 standard) calibration points and for AP2E of 150 (AO1 standard) and 148 (FP5 standard) calibration points. From this dataset, we only keep the calibration points successfully performed on both standards and also add the two sigma filtering on the humidity signal. This gives for Picarro 146 calibration points and for AP2E 138 calibration points for both standards, which are now annotated in table 1.

We acknowledge that the choices we made to show the calibration points were not well suited. Indeed, it gave the impression that the number of Picarro points are much higher than for AP2E. This is due to the fact that the Picarro curves were placed on top, making many of the AP2E points invisible. In addition, the error bars reduced visibility and accentuated this impression, while providing no particular information. We have therefore decided to remove the error bars, indicate the average instantaneous noise over the whole series in the figure legend and improve the display of the curves.

Lines 190 further. This humidity / mixing ratio dependence is widely observed, not only for water vapour measurements, but also for isotopes in atmospheric CO2. In that field, two ways of dealing with it are in use: the 'ratio method', so building ratios first, and then correct for mixing ratio dependence (this is what you do here), the other method is the isotopologue method, which would first analyze the different isotopologues as different species, calibrating first them, and only then build ratios. See for instance papers:

Flores, et al. Calibration Strategies for FT-IR and Other Isotope Ratio Infrared Spectrometer Instruments for Accurate delta C-13 and delta O-18 Measurements of CO2 in Air. Analytical Chemistry {89}, {3648-3655} (2017).

Steur et al. Simultaneous measurement of δ 13C, δ 18O and δ 17O of atmospheric CO2 – performance assessment of a dual-laser absorption spectrometer. Atmos Meas Tech 14, 4279–4304 (2021).

Both have their pros and cons.

Would that be something to try out here? Or at least to mention and discuss. Or is it not applicable at all in this context?

Thank you for this useful comment. We now add citations, at the beginning of this section, of different application which use the mixing ratio dependency correction including the paper from Flores et al., and we specified that we are using the "ratio method" for our study:

We present in this section the characterisation referred in the litterature as the mixing ratio dependency, which is used in various atmospheric isotopic measurements such as O2 (Piel et al., 2024), CO2 (Flores et al., 2017) or H2O (Weng et al., 2020). We use in this study the

most common, "ratio method", which consists in calculating first isotopic ratios from the measured optical spectrum, and then correcting it from the mixing ratio dependency.

To the best of our knowledge, no similar comparison has been performed in the literature for water vapor isotopes. However, we think that the water mixing ratio calibration section is already long (2 pages out of a 13-page article) and will not develop this aspect further for the sake of readability.

FIgure 8 : The Allan deviation is only part of the final uncertainty: the humidity correction and the calibration error also contribute. While that is less important for tracking the diurnal cycle, the calibration error will influence the accuracy of the seasonal cycle.

This is true and very important to have in mind when looking at absolute isotopic values, although calibration errors are complicated to quantify. We've added a discussion mentioning this in section 3.2, in the paragraph "Humidity and isotopic composition dependency". What we want to emphasize with fig. 8 are the OF-CEAS limitations related to the precision of our signal. This characterization shows the minimal measurable water mixing ratio without taking into account error and biases introduced during the calibration procedure.

Line 300 see above: the humidity dependence of dD is relatively speaking larger than that of d18O, and also it calibration uncertainty (fig 7) is, relatively, larger. Would that influence your conclusion?

The uncertainties have to be compared to the diurnal (or seasonal) cycle, which has an 8-times large span for dD compared to d18O. This means that the absolute uncertainty on the dD signal can be 8 times larger to allow equivalent "results" in terms of diurnal cycle interpretation. If we look at the maximal range between the two standard calibration curves (fig. 6, left panel), ie at 50 ppm, we find a difference of 8 permil for d18O and of 80 permil for dD, resulting indeed in a slightly larger difference for dD even if we consider the factor of 8. Concerning the uncertainty from Fig.7, we stay below the factor of 8 for dD compared to d18O, which allows us to consider both isotopes with the same level of confidence.

Finally, we observe that the high humidity dependency is lower for dD, which makes it interesting for interpretation in the high humidity regime, and would provide less calibration errors.

Line 342 "applying a drying on the humidity generator"  ?? unclear to me what you mean.

Thank you for highlighting this, this was indeed unclear. This has been changed by :

"sending dry air through the humidity generator chambers and tubings"

Lines 355-365 (and figure 6) I agree with your conclusion that while the gradual, linear dependence is indeed caused by a spectral 'misfit' (probably indeed the interference of the very strong lines further in the spectrum, or methane?),  the low humidity part strongly indicates sample-to-sample memory effects. This is further supported by the fact

that the depleted ref goes up, and the 'enriched' one goes down.

This effect will not fully vanish in the station, because (1) the residual water vapour is probably quite fractionated, and will thus still be different from outside humidity, and (2) you must calibrate your instrument regularly using two waters with very different isotope values.

I would like to see more discussion of these low-humidity part effects (after all, that is the truly innovative part of the instrument and your paper): for example, are the values (in fig 6) influenced by the 'sample history' before one of the reference waters and how would that be different in the station? Would you expect that sample water vapour with isotope values intermediate between the 'high' and 'low' ref waters scale linearly between them (as you suggest in lines 250-253)? What is the added uncertainty in this region?

We agree with you that the memory effect will not fully vanish, although it will be lower in the stations. In fact, we think that this effect is mainly driven by atmospheric, fractionated water vapor sticking to the tubings: calibration represents less than 5% of the measurement time and most of the time the tubings are exposed to lab air.

Indeed, the fact that the "low ref goes up" and "high ref goes down" indicates that an "enriched" residual water could be responsible for this low humidity divergence, possibly originating from the environmental isotopic composition in the lab. To check this hypothesis, we corrected the calibration curves by assuming a mix with a residual water with a concentration situated between 10 to 20 ppm and an isotopic composition estimated by sending dry air in the instrument through the same tubings and without any water injection. This correction effectively flattened both d18O curves (fig. 6, top left), but not the dD curves (fig 6, bottom left) which do not show any symmetry. This indicates for us that residual water can not be the only source of this low humidity divergence, and that other phenomena, like spectroscopic effects could be responsible for this. This would need a complete and more specific study which is beyond the scope of this paper. We thank the reviewer for having raised this question, and think that a next study focused on this particular question could bring a lot to the community of atmospheric water isotopes.

Here are the answers to your questions:

- concerning sample history: To reduce at maximum this history, during the lab humidity-isotope calibration we always start with the high humidity injections (1000 ppm) and finish with the low humidity step (50 ppm). This means that when the low humidity step is performed, the tubings have been flushed with the same standard for at least 10-15 hours. The 6 calibrations performed across 3 months show no particular trend, which gives us a good confidence on the repeatability of the calibration. No difference was observed by switching the order of the calibration standard.
- concerning the linearity and uncertainty between the low and high ref : in Figure 7, the isotopic compositions measured with TD3 (with corresponding humidities lying in the divergence area, from 67 to 700 ppm) have been corrected using a linear

combination of the highly and lightly depleted standard calibration curve, resulting in a linear dependency between the measured and true value, which supports (at least at first order) the hypothesis of a linear scaling between the calibration curves. The accuracy and precision are given in section 2.3: this uncertainty integrates the instrumental drift over a few months of calibration and the deviation from linearity that one could expect between the high and low ref (indeed, Weng et al. suggested a quadratic relation between the a, b and c parameters used for the mixing ratio dependency fitting)

We are currently including some of the most important elements of this discussion in the manuscript in section 3.2, under the second paragraph renamed "Humidity and isotopic composition dependency and calibration uncertainty".

Line 364-365: 'We insist thus on the importance of calibrating the instrument in the field to correct for those artefacts (Casado et al., 2016).' Indeed! May be change the word 'insist' into 'emphasize' ?

Thank you for this suggestion, the word has been changed.

line 371-372 "The internal architecture of these analysers therefore reduces the risk of breakdowns during the deployment, but requires an expertise to finely tune them. "

To me this feels like a blessing in disguise, or may be just the exact opposite. Deploying these instruments thus always requires expertise (and time and some equipment) in the field. Any more comments to that? For example negative field experience with fixed mounted equipment such as the Picarro's ?

Thank you for this question. The paragraph "interesting features for field operation" has been modified and Picarro was clearly mentioned to avoid any ambiguity (see answer to "Reviewer 3" for more information).

I agree that working with these instruments always requires qualified staff, and it's clear that Picarro devices have the advantage of being more user-friendly, enabling deployment without any specific expertise (e.g. skills for optical alignment). The downside of this ease of use is that in the event of a breakdown, the system is not sufficiently open, making it impossible for users to intervene "in depth" in the device. Since remote intervention via remote desktop is complicated in Antarctica, this would mean sending the instrumentation back to mainland France for repair, and thus abandoning the mission.

"Competing interests. The authors declare that they have no conflict of interest."

Two of the co-authors work with the (commercial) producer of the instrument. How do they avoid a conflict of interest?

Indeed, thank you for this remark, this work has been performed in collaboration with the instrument producer. We changed the competing interest section with this new paragraph:

This work was made possible by a collaboration with the company AP2E who produced the analysers presented in this manuscript. The characterisation of the analysers was done independently at LSCE, with no interference from AP2E.

---

## Author Comment (AC2)

**Reviewer 3 - Anonymous**

The manuscript titled "OF-CEAS laser spectroscopy to measure water isotopes in dry environments: example of application in Antarctic" by Lauwers et al. is a much needed study on a promising technique to measure water vapor isotope composition at very low humidity. The manuscript includes all of the information one would hope for to employ their method and requires only minor revisions before publication. Indeed, the authors would go a long way towards publication readiness by clarifying the text and refining the grammar.

We thank the reviewer for his careful reading and recommendations. The answers are inserted in red and the citations from the manuscript are indicated in blue.

General comments for clarity:

- Make the figures and tables, including their captions, more independent of the main text. Telling a reader the same information twice (e.g., in the figure and in the main text) helps with clarity.

Thank you for this comment, we are adding all the information needed to understand the figures, both in the main text and in the figure captions.

Instrumentation description:

- I expect this study and instrumentation to be popular among water isotope laboratories and researchers. As such, please describe the specific model of AP2E used. Or, if all three are custom, please indicate so. Consider including a brief description of how these particular OF-CEAS instruments differ from those in Casado et al., 2016. Are they the same?

The two OF-CEAS instruments presented in this study correspond to the Proceas © model, while the first instrument deployed by Casado et al. in 2016 was a prototype assembled at the LIPhy laboratory (Grenoble). The technique compared to the laboratory prototype remains the same, but with improvements in terms of temperature regulation, pressure regulation and robustness (electronics, optomechanics). We added a paragraph in the manuscript at the beginning of section 1.1 to explain the specificity of this new model.

- Use of the description "V-shape" does not help the reader understand the instrument. Perhaps it would be more informative to say something about the lack of fiber optic cabling in favor of highly reflective mirrors?

The reason why the cavity has a V-shape is indeed not explained, because that would be beyond the scope of our paper and has been already been documented in the literature. We feel it necessary to be accurate, at least in the introduction, by stating that the OF-CEAS cavity uses a V-shaped cavity, as otherwise efficient optical feedback would not be possible. The V-shape cavity is essential to cancel direct reflection and only send back to

the laser the resonating light, as the incident light is injected with an angle. Additionally, the V-shape enables it to reach twice the length of a normal linear cavity and increase the laser light/gas interaction length while keeping a compact size (40 cm long).

We added the reference to chapters 1 and 5 of Romanini and Morville 2014 in the introduction, which gives a detailed description of the OF-CEAS principle, including the principle of the V-shape cavity. In subsection 1.1, we added a paragraph to introduce the elements discussed here, and cited Clement Piel et al. 2024 who describe in detail the OF-CEAS set-up such as implemented by the AP2E company.

2.1.2 Long-term stability at Dumont d'Urville station:

   - Do the authors expect the variable water vapor concentration requiring "An additional filtering has been applied to remove the points with a non- stable humidity, i.e. with a humidity value standing outside the 2-$\sigma$ interval." is due to the LHLG unit or to the OF-CEAS instrument itself?

The non-stable humidity is due to the LHLG unit. The absolute humidity traces measured by both CRDS (Picarro) and OF-CEAS techniques appear to be nicely superposed. Moreover, the precision for the absolute humidity on these analysers allows us to clearly observe the LHLG humidity overshoots and instabilities, while we are looking at signals that are 2 to 3 orders of magnitude smaller when measuring isotopic ratios.

Figure 5:

   - De-emphasize the green data by making it a lighter shade and placing them behind the OF-CEAS data.

   - I suggest expressing the y-axes as residuals in the same way the authors have already done with Figs 3 and 6. The wide range that must necessarily be used when expressing the values in their absolute terms may hide drift or biases.

Thank you for this useful remark. For a better clarity of the figure, we centered the y-axis around the mean isotopic value, with the same scale between the left panel (AO1 standard) and right panel (FP5 standard). We removed the error bars which did not provide very useful information and overloaded the figure. We also added the standard deviation of the whole series. The numerical values are displayed in Table 1 to avoid overloading of figure 5.

   - The caption sentence "Each point represents the average of the final minimum 5 minutes of 1000 ppm humidity plateaus and the error bars correspond to the associated standard deviation" is unclear. Does "minimum" imply that sometimes it is longer? Be specific.

Thank you for your comment. Ideally, we take a 10-minute window for the drift calibration value, which ensures a stable isotopic and humidity value. However, it can happen that the plateau is not completely stabilized over the last 10 minutes (or that the

occurrence of an air bubble generated by the calibration instrument creates instabilities), so in this case we reduce the time window with a lower limit that we set at 5 min. The caption has been modified accordingly.

- Why would "plateaus" exist with this particular dataset? The caption currently reads as if all of these data were collected at 1000 ppm. Please clarify.

The value of 1000 ppm corresponds to the theoretical value delivered by the LHLG with the corresponding settings (water and air flux, pressure, etc.). In reality, the generated humidity is affected by instabilities of the water flux and varies from one calibration to another, leading to a typical standard deviation of ~ 30 ppm of the humidity over the presented calibration dataset in fig. 5. These small variations of the humidity are negligible in terms of impact on humidity dependency correction of the isotopic value. So the measured d18O and dD signals during the calibrations do not differ from what would have been obtained at exactly 1000 ppm. We are clarifying this aspect in section 2.1.2.

Table 1:

- If these variance estimates are from the data presented in Fig 5, I suggest deleting this table and stating these values in the Fig 5 caption text. If the authors choose to keep the table, state in the table caption that the variance estimates are from the data presented in Fig 5. If they are not from the data in Fig 5, please describe their origin.

Indeed, Table 1 gives the standard deviation calculated from the time series plotted on Figure 5. As indicated in a previous answer concerning Fig. 5 (referee 1), we decided to indicate the standard deviation in a graphical way to avoid overloading the figure, and specify the exact numerical value on this table. This will be stated in the caption.

Figure 6:

- Please describe in the caption what is meant by "ref" in the y-axis titles.

We added this sentence in the caption:

All curves are referenced to the isotopic composition measured at 500 ppm (left panel) and 1000 ppm (right panel), denoted "ref" on the y-axis.

Table 2:

- Does "lightly depleted" and "highly depleted" refer to standards shown in Fig 6? Please describe in the table caption what is meant, or better yet, remove the ambiguous terms in favor of the actual reference water names.

We added the actual standard names which we use as internal references, i.e. AO1 for the lightly depleted standard and FP5 for the highly depleted standard.

Figure 8:

- The caption is confusing. The authors are not plotting Allan deviation after 24 hours, they are plotting the predicted standard deviation after 24 hours of integration as predicted from an Allan variance analysis.

- The wording describing the dotted lines is confusing. It currently sounds as if they are the predicted range one would expect during the course of a 24 hour period. The main text, however, suggests these dotted lines are the authors desired "noise threshold".

- Is this threshold one standard deviation or perhaps a 95 % confidence interval? Please reword the caption.

- Make the right side axis title and tick labels red, in the same way the authors made the left side blue.

- Lastly, the font size of the stations on the map need to be much larger.

Thank you for these comments which will help make the figure more readable and less confusing. The figure 8 caption has been changed to correctly describe the Allan deviation and the threshold. The threshold corresponds to the upper limit for the standard deviation as predicted from the Allan deviation study.

Here is the new caption:

"On the left, histogram representing the year fraction (expressed in %) below a fixed humidity content for 4 sites situated along the transect. For each humidity, we plotted the associated standard deviation after 24 hours of integration $\sigma(\delta i)$ as predicted by the Allan deviation study for $\delta^{18}O$ and $\delta D$. The dotted horizontal lines represent the $\delta^{18}O$ (red) and $\delta D$ (dark red) upper thresholds for the standard deviation to confidently study the diurnal cycle. These thresholds are set at a value 10 times lower than the amplitude of the diurnal cycle, to ensure a correct signal resolution (see discussion in the text). On the right, map with the location of the 4 instrumented sites for the AWACA deployment."

The right side axis, title and tick labels are now in red, the size of the font for the stations is larger.

Section 3.1

- This section would be better positioned after the discussion since it is a proposed application of what the authors intend to do after having completed the current study.

Thank you for this comment. Indeed, the proposed structure for section 3 was in the original manuscript and seemed more obvious at first sight. But during the co-authors revisions, we found that a number of elements discussed in the current section 3.2 were not clearly introduced, which caused some repetitions in the manuscript and a loss of clarity. For example, in line 327 we use the result plotted in fig. 8 from the previous section. Again, in line 341, among the proposed solutions for improving the

measurements, we discuss the calibration strategy, which was introduced in the previous section.

The discussion thus appeared more grounded when presenting first a concrete example of application and the logic of section 3 is the following: in subsection 3.1, the results of section 2 are discussed and summarized (Fig. 8) to propose a calibration scheme for an example of application in Antarctica (AWACA project). With this specific application in mind, the aim of subsection 3.2 is to give the limits and advantages of the OF-CEAS instrument compared to the state-of-art Picarro CRDS, and finally give some perspectives of improvement. In subsection 3.2, we propose for example a number of ways to improve the annual coverage of water vapor isotopes (shown on Fig. 8), such as increasing the calibration frequency. We also discuss in the last paragraph the advantage of using OF-CEAS instrumentation in the field for the specific case of autonomous operation in remote sites, such as in the AWACA project.

Figure 9:

   - Similar to my comment for Fig 8, reword "The noise is obtained from short-term Allan deviations at $\tau$ = 2 min..." to something like "Standard deviation is predicted from an allan variance analysis for the 2 minute integration time...".

Absolutely, the caption has been changed.

Discussion Line 325:

   - The sentence "It shows an optimum stability range of ~ 15 min, followed by a drift period in the hour range and finally a stabilisation of the signal in the day range" is confusing. Are the authors referring to the #1169 instrument at 500 ppm $H_2O$ and d18O? The blue series is the only series that does seem to be achieving stability in the day range. All others continue to drift or have insufficient data to make any conclusions about their trajectory (e.g., dD #1169).

Indeed, the system does not reach a clear stabilisation, but the AD curves clearly show that although the drift does not disappear in the day range, the slope of AD slows down after 10 hours and its value lies in a satisfying range. We are modifying the text accordingly.

Interesting features for field operation:

   - This is an important section and seems to compare OF-CEAS to Picarro's CRDS. The glue, the inability to clean, lack of needed software. I suggest the authors state plainly that AP2E's design and software are more conducive to remote field deployment than Picarro's CRDS.

We believe indeed that for our specific application, the system designed by AP2E is better suited than Picarro, and we modified the paragraph to clearly mention Picarro's CRDS. But this has to be put into perspective: in the case of water vapor isotope

observatories at mid-latitudes where the humidity is not so low, there is an interest in Picarro systems which are particularly adapted: they are very robust, more "plug-and-play" and show higher performance that what would be obtained with AP2E OF-CEAS technique.

---

## Author Response (AR2)

Dear Authors,

I would like to thank you very much again for submission to AMT. Your manuscript underwent a thorough review process and both referees attest to the high quality of your revised submission. However, as pointed out in one of the reports, the grammar needs to be improved substantially. I therefore decide to publish subject to minor revisions (review by editor) and I ask you to consider all of the suggestions provided by the corresponding referee report as well as the comments that are detailed further below. I also have three minor topical questions that should be addressed in the revision.

1. Figure 2, which has been the subject of previous referee comments should be improved. At first sight it seems that scales of the axes on the right- and left-hand side are quite different, but as a matter of fact they are very similar. I therefore suggest that you improve readability of the graph by using units of 1E-10 1/cm on the right and on the left hand side and plot (absorption - 3164) E-10 1/cm on the left instead of just the absorption signal. To ease the mind of the spectroscopically aware reader and especially to allow quick location of eventual traces of the H218O peak, please also provide a second x-scale (x-axis on top) in wavenumber units. Finally, it is not immediately clear how the residual trace depicted in Fig. 2 has been obtained. Has the fit been applied directly to the absorption signal shown in the figure or to a background corrected version of the spectrum, and which fitting function has been used (Voigt profile, what about the empty cell signal, methane, baseline function, ... ) ?

We thank the editor for this comment which helped us to improve the readability of Figure 2. The left axis has now units of 1E-10/cm and the background absorption offset has been subtracted. This offset corresponds to the absorption losses inside the optical cavity (e.g by light scattering on the cavity walls). We also added a secondary x axis with the wavenumber.

[Figure]

The fit has been applied directly to the absorption signal shown in the figure with Voigt profiles for the water and methane absorption lines, and the background absorption (the baseline) is fitted with a quadratic function.
A sentence has been added (line 86 in the marked-up manuscript): "The spectral fitting is performed using Voigt profiles for the water and methane absorption lines and an additional quadratic baseline to account for background absorption losses."

2. The absolute numbers for the 2-day stability are certainly not critical for the outcome and conclusions of the paper, but the last data points of all four blue and red curves in Fig. 4 are likely artifacts and need to be verified and/or removed. At least, these points don't seem to be produced by a conventional Allan deviation analysis, which provides data points that are equally spaced on a logarithmic scale (at τ = Δt 2^n, where Δt = 8 000 s and n = 0, 1, 2, …). While points at 8 000 s, 16 000 s, 32 000 s, 64 000 s and 128 000 s are shown correctly in Figure 4, the next point should be at 256 000 s, but it is at 172 800 s! Moreover, and this is even more puzzling on Figure 4, the y-values at t = 2 days are identical to the y-values at 128 000 s. This points to a critical problem in the analysis software or the production of the graph. Therefore, I need to ask you to carefully check your data using independently verified and freely available software, such as DATAPLOT from NIST or the avar function provided in the statistical software R. These software packages also provide error bars which for long integration times are currently missing in the graph.
On a related issue, when mentioning the stability over two days (lines 161-162), it would be more appropriate to use the maximum of the deviation curves between 10000 and 100000 s. Please also add "P. Werle, R. Mücke, and F. Slemr. The limits of signal averaging in atmospheric trace-gas monitoring by tunable diode-laser absorption spectroscopy (TDLAS). Appl. Phys. B, 57:131–139, 1993" to your list of references. These authors have brought the Allan variance stability analysis to the domain of environmental spectroscopy.

We thank the editor for pointing out several aspects of this figure and the error on the last point of the curve at t = 2 days, which is now removed. We propose a new way of presenting the Allan deviation of the 100 ppm and 500 ppm calibrations. This should help to clarify the figure and answer the questions

[Figure]

- The long-term AD including "the gaps" and starting at t = 8000 s is now showed with empty markers to be clearly distinguished from the more classic AD determination (without gaps). There is a poor confidence on the last point of this AD so we decided to remove it.

- From τ = 8000 s, the points corresponding to AD without gaps (red and blue filled markers) are overlapping with the long-term AD points within the uncertainty ; to avoid any confusion (particularly concerning spacing between points) we no longer plot them in Figure 4.
- This Allan deviation has been compared to adev and oadev function available in the Python package "AllanTools" (see https://allantools.readthedocs.io/en/latest) which provide the same results.
- The maximum values of AD between 10000 and 100000 s have been added in the text (line 174 and 175 of the marked-up revised manuscript).

Thanks for the reference suggestion. It has been added (line 108 of the marked-up revised manuscript)

3. Five different standards are used. It would be convenient to summarise their isotope composition in one Table.

A table has been added :

|  | $\delta^{18}O$ | $\delta D$ |
|---|---|---|
| Ross 7 | (-18.94 ± 0.05) ‰ | (-146.0 ± 0.7) ‰ |
| AO1 | (-30.60 ± 0.05) ‰ | (-238.3 ± 0.7) ‰ |
| TD3 | (-40.19 ± 0.05) ‰ | (-313.6 ± 0.7) ‰ |
| FP5 | (-50.52 ± 0.05) ‰ | (-394.7 ± 0.7) ‰ |
| OC4 | (-53.93 ± 0.05) ‰ | (-422.7 ± 0.7) ‰ |

Table 1: List of in-house standards used in this study and their SMOW/SLAP calibrated $\delta^{18}O$ and $\delta D$ values (determined with a Picarro analyser for $\delta D$ and mass spectrometry for $\delta^{18}O$).

Wording & grammar suggestions

(in the following, the abbreviation 'l.' is used to indicate a line number)

We thank the editor for this careful reading improving the quality of the manuscript. The line numbers below correspond to the marked-up version of the revised manuscript.

General comment: The use of commas and points in decimal numbers is confusing. For example, sometimes a comma is used to separate three digits (line 142 '8,000 s', l. 247 '50-1,500 ppm' ), sometimes it is not (eg. line 162 '1000 ppm', l. 241 'above 1000 ppm', l 264 '50 to 6500 ppm', etc. ). This even occurs on the same page of the manuscript and triggers the question whether the comma is a decimal comma or not. The general recommendation for scientific publications (see BIPM, IUPAC, etc.) is that thin spaces can be used to group digits and that 'neither dots nor commas are ever inserted in the spaces between groups'. I would like to ask you to comply with this convention. We removed all the digits separators; thin spaces have been inserted

l. 19-20 Please revise the last phrase 'The high finesse instrument demonstrates a stability up to two days of acquisition with a limit of detection down to 10 ppmv humidity for $\delta$D and 100 ppmv for $\delta$18O.' Please rephrase. Information given in the abstract should be concise and clear, but here one can only guess what a 2 day stability means if one has read the article and the numbers for the detection limits ($\delta$D, $\delta$18O) are missing entirely.

The last sentence has been rephrased and now reads : "With a drift calibration every 48 hours, the stability demonstrated by the high finesse instrument allows to study diurnal cycles down to 10 ppmv humidity for δD and 100 ppmv for δ$^{18}$O."

l. 28 delete 'such'.
done

l. 31 'The CRDS method is commonly implemented by Picarro company, and gives a high stability through the measurement of the photon lifetime inside the optical cavity instead of the direct absorbed light.' If the main intention is to inform the reader on the physical principles, I suggest to write 'The CRDS method, which is commonly implemented by Picarro, achieves a high measurement stability through the ....'
done

l. 53 'This technique was first implemented for water vapour isotope analysis with a laboratory prototype (Landsberg et al., 2014) but never successfully deployed in the field, with stable working conditions.' -> 'This technique was first implemented for water vapour isotope analysis using a laboratory prototype under stable working conditions (Landsberg et al., 2014), but never successfully deployed in the field.

Sentence rephrased as : " This technique was first implemented for water vapour isotope analysis with a laboratory prototype under stable working conditions (Landsberg et al., 2014), but never successfully deployed in the field for extended periods.

l. 54 Start phrase with 'In this paper, we present'
done

l. 67 Cite in text 'For a complete description of the ProCeas® system, the reader may refer to the recent article of Piel et al. describing the OF-CEAS spectrometer used for atmospheric O2 isotopes measurement (Piel et al., 2024).' -> 'For a complete description of the ProCeas® system, the reader may refer to the recent article of Piel et al. (2024) describing the OF-CEAS spectrometer used for atmospheric O2 isotopic measurement.'
done

l. 69 'Water isotopes OF-CEAS spectrometers use ...' -> 'The OF-CEAS spectrometers for the measurements of water isotopologues use ...'
done

l. 74 'retrieved from HITRAN database' -> 'calculated from the HITRAN 2020 database'
done

l. 78 'is performed' -> 'is achieved'
done

l. 81 'the shown spectrum' -> 'the spectrum'
done

l. 82 'Nitrogen' -> 'nitrogen'
done

l. 86 'in a various range of gas matrices (pure nitrogen, atmospheric dry air and finally synthetic air with a low water content).' -> 'for a wide range of different gas matrices (pure nitrogen, atmospheric dry air and finally synthetic air with a low water content).'
done

The titles of Figs 1 and 2 should be deleted. The second seems to be inappropriate as it is essentially the spectrum of a very dry cavity.
Titles have been deleted

l. 94 'in 600 ms' -> 'within 600 ms'
done

l. 95 'correct' -> 'useful'
done

l. 96 'In order to keep a fast, real time data acquisition, the fitting algorithm is tuned so that most parameters are fixed.' -> 'In order to keep the data acquisition fast and in real time, the fitting algorithm is tuned by fixing most parameters.'
done

l. 113 'maximal' -> 'maximum'
done

l. 117 Start phrase with 'In this section we present'
done

l. 125 'from few hours' -> 'from a few hours'
done

l. 139 'two first' -> 'first two'
done

l. 143 8,000s -> '8000 s
done

l. 155 Use 'tau' instead of 't' in the white noise law.
done

l. 159 'arising along the laser to cavity optical path' -> 'arising along the optical path between laser and cavity'
done

l. 161 'After two days, we calculate an AD ...' -> 'At a delay of two days, we observe an AD ...'
done

l. 164 'hourly region' -> 'on the time scale of a few hours'
done

l. 177 Write 'In Figure 5, we present ...'
done

l. 200 'Indeed, large temperature variation' -> 'Indeed, large temperature variations'
done

l. 205 Replace 'lightly' by 'slightly' in Table 1, I likewise recommend to replace 'highly depleted' by 'strongly depleted'
done, and elsewhere in the manuscript as well

l. 235 Start phrase with 'In Figure 6, we show'
done

L. 223 - 225 Use italics for mathematical symbols f, h.

l. 258 Avoid using the comma as a separator for
done

l. 271 'further away than the large water absorption peak' -> 'further away from the large water absorption peak'
done

l. 279 'linear, global' -> 'global linear'
done

l. 288 'inside' -> 'within'
done

l. 327 'we plotted the associated standard deviation after 24 hours $\sigma(\delta_i)$ as predicted by the Allan deviation study' -> 'we plotted the Allan deviation for 24 hours $\sigma(24h)$ as predicted by the study in section xxx' If you want to indicate the delta values, use the subscript notation as in Fig. 4.
&
l. 329 An Allan deviation is not a standard deviation: 'standard deviation' -> 'Allan deviation'. Please make according corrections also in the manuscript.

For these two comments, in fact we have modified the first manuscript to follow the request of one reviewer (reviewer 3) : " - The caption is confusing. The authors are not plotting Allan deviation after 24 hours, they are plotting the predicted standard deviation after 24 hours of integration as predicted from an Allan variance analysis."
So we feel quite uncomfortable changing now, but we can discuss it further

l. 328 use superscript for 18
done

l. 335 'This threshold value is indicated on the figure by the horizontal dotted line' -> 'These threshold values are indicated on the figure by the horizontal dashed lines''
done

l. 340 'led' -> 'leads'
done

l. 330 'correct' -> 'proper'
done

l. 338 'that we should preferably consider $\delta$D to $\delta$18O in very dry environments.' -> 'that we should prefer acquisitions of $\delta$D over measurements of $\delta$18O in very dry environments.'

done

l. 342 'showed' -> 'shows'
done

l. 355 'We compare in Figure' -> 'In Figure , we compare'
done

l. 359 'This shows the ability for the OF-CEAS technique to capture with a high precision transient events' -> 'This shows the ability of the OF-CEAS technique to capture transient events at high precision'
done

l. 369 'on the Allan' -> 'by the Allan'
done

l. 376 'This leads for the OF-CEAS' -> 'This leads'
done

l. 394 replace by a more appropriate term the word 'global', or delete it.
removed

l. 395 'The low humidity divergence' -> 'The divergence at low humidity'
done

l. 398 'to changing gas mixture can' -> 'to a changing gas mixture can'
done

l. 412 'and in particular possible methane contribution (see Figure 1) ...' -> 'and in particular a possible contribution from methane (see Figure 1) ...'
done

l. 416 'those'-> 'these'
done

l. 416 'Such artefacts are complicated to evaluate ...' -> 'Such artifacts are difficult to evaluate ...'
done; we kept artefacts as more in line with GB English like "vapour" (see your recommendation below)

l. 422 'vapor'->'vapour', please check all instances
done and checked

l. 423 'great interest for' -> 'great interest in'
done

l. 438 'through long-term' -> 'through a long-term'
done

l. 446 'like in-situ measurement of the isotopic composition of individual snowflakes.' -> 'like the in-situ measurement of the isotopic composition of individual snowflakes.'
done

l. 449 'The OF-CEAS analyser limitations highlighted in this article concern the instabilities encountered in the hourly range, which we attribute to parasitic interferences.'-> 'The OF-CEAS

analyser limitations highlighted in this article are instabilities that develop at the time scale of a quarter-hour or so, which we tentatively attribute to the evolution of parasitic interferences.'
Done

I think the authors have adequately responded to my previous review, except for improving the grammar. I think the manuscript will be ready for publication after incorporating the below edits or completing a thorough proofread themselves. The below suggested clerical edits use line numbers from the track-changes document with file name "egusphere-2024-2149-ATC1.pdf".

We would like to thank the reviewer for this thorough and careful edit work.

line 22 - "Stable water vapour isotopes" should read "Water vapour stable isotope", and note 'isotope' is singular in this case
DONE

line 31 - either change "by Picarro company" to read "by the Picarro company" or "by Picarro". Below, on line 55, you have "with the AP2E company". Be consistent.
DONE and whole manuscript checked

line 33 - "stable water vapour isotopes" should read "water vapour stable isotopes"; check throughout manuscript for this change; you are measuring the stable isotopes in water vapour, not the isotopes in stable water vapour.
DONE and whole manuscript checked

add comma after "e.g." throughout manuscript; you are saying "for example, cold fronts, cyclones"
DONE and whole manuscript checked

line 36 - change "on board of boats" to read "on board boats"
DONE

line 37 - change "number of studies is also" to read "number of studies are also"
DONE

line 41 - change "isotopic records in ice core which" to read "isotopic records in ice cores, which"
DONE

line 43 - change "However, CRDS struggle" to read "However, CRDS struggles"
DONE

line 45 - change "or in altitude" to read "or at altitude"
DONE

line 48 - change ", allowing to narrow down" to read ", to narrow down"
And
line 49 - change "the cavity resonances (Morville" to read "the cavity resonance (Morville" OR if you wish to keep resonances plural, then change "laser emission frequency by locking it to the cavity resonances" to read "laser emission frequencies by locking them to the cavity resonances". The plural form is consistent with the following sentences.
We have rephrased this sentence to make it clearer : "This allows us to stabilise the laser emission frequency by locking it successively to the multiple cavity resonances (Morville et al., 2014; Romanini et al., 2014)."

line 58 - change "water vapour isotopes monitoring" to read "water vapour isotope monitoring"
DONE

line 65 - change "in term of robustness" to read "in terms of robustness"
DONE

line 68 - change "atmospheric O2 isotopes measurement " to read "atmospheric O2 isotopic measurement "
DONE

line 69 - change "Water isotopes OF-CEAS" to read "Water isotope OF-CEAS"
sentence modified following the suggestion proposed by the editor —> "The OF-CEAS spectrometers for the measurement of water isotopologues "

line 74 - change "from HITRAN database" to read "from the HITRAN database"
DONE

Figure 1 minor suggestion - place the dotted red sum line behind the H2O blue line.
We prefer to leave the Figure as it is, since if we place the dotted line behind the blue line, it will no longer be visible.

Figure 2 caption - change the comma at the end of "than 52 ms," to a period.
DONE

line 97 - change "time is of 52 ms in" to "time is 52 ms in" or "time is approximately 52 ms in"
DONE

lines 109, 110 - change "gives an easy access" to "gives easy access"
DONE

line 114 - change "This permits to assess the spectrometer" to read "This permits the spectrometer to assess"

We have changed to "This allows the assessment of the spectrometer stability"

line 117 - change "by AP2E company." to read "by AP2E."
We changed to read "by the AP2E company" for consistency with previous corrections (your suggestion for line 31)

line 124, 125 - change "perform Allan deviations (AD) measurements" to read "perform Allan deviation (AD) measurements"
DONE

line 148 - reword "The use of the second dataset enables to reach a time range" to read "Using the second dataset allows for a time range"
DONE

line 157 - This sentence is rather confusing as structured; consider rewording it to make the comparison more apparent: "The ADs of the low-humidity analyser follow a white noise decay during several minutes, with a minimal value of 0.1 ‰ (resp. 0.06 ‰) for δ18O at 100 ppm (resp. 500 ppm) and 0.5 ‰ (resp. 0.2 ‰) for δD at 100 ppm (resp. 500 ppm)."

The reworded sentence reads "The ADs of the low-humidity analyser follow a white noise decay during several minutes, with a minimal value for $\delta^{18}O$ of 0.1 ‰ at 100 ppm and 0.06 ‰ at 500 ppm (0.5 ‰ and 0.2 ‰ for δD at 100 ppm and 500 ppm, respectively)"

line 161 - Similarly, this sentence has the same confusing comparison structure: "After two days, we calculate an AD of 1 ‰ (resp. 0.09 ‰) for δ18O at 100 ppm (resp. 500 ppm) and of 2.5 ‰ (resp. 0.7 ‰) for δD at 100 ppm (resp. 500 ppm)."

Similar rewording done

line 174 - change "two standards injections" to read "two standard injected"

DONE

line 177 - change "performed all over" to read "performed over"

DONE

line 182 - change "After a filtering" to read "After filtering"

DONE

line 184 - change "calibrations for the CRDS analyser" to read "calibrations for the CRDS analysers"

the corrected sentence reads "we obtain 138 calibrations for the OF-CEAS analyser and 146 calibrations for the CRDS analyser"

Figure 5 caption, line 194 - the sentence starting with "The blue line (resp. green line)" is confusing as worded similar to the above line 157, and 161 comments.

The sentence has been reworded and reads "The blue line corresponds to the AP2E OF-CEAS dataset smoothed over a 5-point window, and similarly the green line corresponds to the smoothed Picarro CRDS dataset.

line 199 - change "that the analysers calibrations" to read "that the analysers' calibrations"

DONE

line 201 - change "variation have been registered" to read "variation has been registered"

changed to "variations have been registered"

line 202 - change "This points out the need for a" to read "This underscores the need for a"

DONE

line 212 - change "such as spectroscopic effect affecting the fitting procedure, or memory effect)" to read "such as spectroscopic fitting or memory)"

We feel that this suggested wording is too abbreviated and would prefer to keep the original sentence.

line 214 - remove "contents"

DONE

line 221 - change "using the additionnal calibrated" to read "using the additional calibrated"

DONE

line 224 - remove "then"
DONE

line 239 - change "various series of measurement" to read "various measurements"
DONE

line 241 - change "and finish with the" to read "and finishes with a"
DONE

line 246 - change "A first humidity sequence" to read "An initial humidity sequence"
DONE

Figure 6 caption, line 264 - change "The y-axis are" to read "The y-axes are"
DONE

line 272 - change "observe a larger noise" to read "observed increased noise"
DONE

line 284 - change "and for longer time period" to read "and for longer time periods"
DONE

line 292 - I believe a negative sign is missing in the phrase "0.7 ‰ for δ18O". Based on Figure 7, TD3 is below the expected value.
Yes, we were thinking in terms of absolute value, but you are right —> corrected

Figure 7 caption, line 297 and elsewhere - You have many notations to express "VSMOW-SLAP". I see " IAEA VSMOW/SLAP" and "VSMOW-calibrated" and "VSMOW-SLAP". Pick one notation and change throughout. I prefer one without "IAEA".
We changed to VSMOW-SLAP in all the document

line 323 - By now I am accustomed to the "(resp. )" structure. However, I still think the authors should consider an alternative. Furthermore, the "(resp. dD)" needs a delta symbol rather than a lower case d.
So we keep the "resp." structure in short sentences only. dD changed to δD

line 328 - superscript the 18 in "δ18O"
DONE

line 332 - change "are of the order of 10 ‰ " to read "are of order 10 ‰ "
DONE

line 340 - change "water vapour isotopes monitoring" to read "water-vapour-isotope monitoring"
DONE

line 355 - I suggest changing "permits to reach extreme levels of precision at low water concentrations" to read "allows for high isotope-ratio precision at low water concentrations"
DONE

line 356 - change "at 2 minutes of the" to read "at 2 minutes integration of the"
DONE

line 359 - change "to capture with a high" to "to capture high"
changed to "to capture transient events at high precision" (editor's suggestion)

line 372 - change "acquisition time – enabled to keeping enough precision to " to read "acquisition time, allowing for enough precision to "

sentence modified

line 375 - change "quantify the water isotopes concentration" to read "quantify the water isotope concentration"

DONE

line 388 - change "and tubings before" to read "and tubing before"

DONE

line 390 - change "over all the year" to read "over a year"

changed to "over the whole year" (to imply the different climatic conditions of the year, and not only the duration)

line 415 - change "We emphasize thus the " to read "We emphasize the "

DONE

line 427 - change "offers also the" to read "offers the"

DONE

line 428 - change "like for example various" to read "like various"

DONE

line 433 - change "spectrometers based on the OF-CEAS" to read "spectrometers using the OF-CEAS"

DONE

line 440 - change "drift on neither instrument" to read "drift on either instrument"

DONE

---

## Author Response (AR3)

Dear Authors,

Thank you for the revised version of your manuscript which takes into account all previously raised questions. Please note that there are still three minor or very minor issues:

Dear Editor,

We deeply thank you for your careful reading

1. Figure 2.
The y-axis label should read "Absorption (10^-10 cm^-1) - 3416" to be dimensionally consistent.

Done

2. Allan Variance
L171 "At a delay of two days, we observe an AD for δ18O of 1 ‰ at 100 ppm and 0.09 ‰ at 500 ppm (2.5 ‰ and 0.7 ‰
for δD at 100 ppm, and 500 ppm, respectively). For comparison, the maximum values of AD for δ18O between 104 and 105 s
are 1 ‰ (100 ppm) and 0.23 ‰ (500 ppm); for δD, 3.9 ‰ (100 ppm) and 1.3 ‰ (500 ppm). "
Since there are no data points at 48 h, nothing can be observed at a delay of two days. The text must be adapted accordingly (l 171, eventually also the text in the abstract). Moreover, the last few points at the end of an Allan Variance analysis are usually not very reliable and the observed levelling-off of the red AD curves might not be valid. If required, a safer way to extrapolate to 2 days would be to assume a linear drift of the instrument (slope = +1/2).
Finally, please show all data in Fig. 4 with error bars (or none). It is mostly the long tau-values that are interesting.

In fact, we had in mind this 48 h "reference" because of the calibration scheme used so far, but with the new improvements to reduce the time between two calibrations, the values at 24 hours are indeed more relevant for the paper and future work.

—> we refer now to the AD values at one day in the text (line 173) and indicate the one-day and two-day dotted lines in the Figure but without giving speculative extrapolated AD values.

We acknowledge the envelopes in Figure 4 were misleading since they were related to the dispersion of the different AD analyses performed only for short time scales. For the homogeneity of the figure, we have removed them.

3. Reference Time
In the abstract, the discussion of the Allan deviation and the discussion of the instrument use in the AWACA project, different reference times are used. While it is 48 h for the former, section 3.1 (and Fig. 8) refers to 24 h. This is confusing. The presentation should either be homogenized or the choice of different reference periods explained better.

It has been homogenized to refer to 24 h as explained above (lines 19; 173 and following)

---

## Author Response (AR4)

**Public justification (visible to the public if the article is accepted and published)**:
Dear Authors,

Thank you for submitting the revised manuscript. Given that you have taken into account all previously requested changes, I am glad to decide that the article be published subject to two technical corrections:

1/ Taking the scaling of the top axis in Fig. 2 as a reference, the frequency decreases from the left to the right. This is conflicting with the FSR (= 188 MHz) being a positive number and the frequency scaling on the bottom given in units of FSR. Please check the scaling of both axes (either the top axis has to be reversed or negative numbers have to be used on the bottom axis).

**In Fig. 2, the optical frequency $\nu$ decreases indeed from the left to the right like the wavenumber $\sigma = \nu/c$, while the wavelength increases from the left to the right (see Fig. 1). Fig. 2 is intended to show the raw spectrum as measured by the instrument, and comes after Fig. 1 which shows the theoretical absorption spectrum. We prefer to keep the same convention (wavelengths increasing from left to right) between the two figures to maintain a consistency in the reading. In Fig. 2, the bottom x-axis corresponds to the resonant mode numbers (each mode being separated by one FSR = 188 MHz) in the chronological order of apparition (this is done by applying a ramp on the laser injected current). To avoid any confusion, we changed the x-axis by "Mode number", which is dimensionless.**

2/ replace 'two days' by 'almost two days' in line 153 of the corrected manuscript.

**Done**

Please take into account the requested changes so that the manuscript can go into the next production stage.
This is also the occasion to thank you again for considering AMT/EGUSPHERE for publication of your scientific work.

With kind regards
Christof Janssen